# Effective Prediction of Prostate Cancer Recurrence through the *IQGAP1* Network

**DOI:** 10.3390/cancers13030430

**Published:** 2021-01-23

**Authors:** Yan Gu, Xiaozeng Lin, Anil Kapoor, Taosha Li, Pierre Major, Damu Tang

**Affiliations:** 1Department of Medicine, McMaster University, Hamilton, ON L8S 4L8, Canada; guy3@mcmaster.ca (Y.G.); linx36@mcmaster.ca (X.L.); 2Urological Cancer Center for Research and Innovation (UCCRI), St Joseph’s Hospital, Hamilton, ON L8N 4A6, Canada; akapoor@mcmaster.ca; 3The Research Institute of St Joe’s Hamilton, St Joseph’s Hospital, Hamilton, ON L8N 4A6, Canada; 4Department of Surgery, McMaster University, Hamilton, ON L8S 4L8, Canada; 5Life-Tech Industry Alliance, Shenzhen 518000, China; litaosha@genomics.cn; 6Department of Oncology, McMaster University, Hamilton, ON L8S 4L8, Canada; majorp@hhsc.ca

**Keywords:** IQGAP1, prostate cancer, prostate cancer recurrence, xenografts, transgenic mouse PC, biomarkers

## Abstract

**Simple Summary:**

Prostate cancer (PC) is a leading cause of cancer death in men in the developed countries. PC mortality is related to tumor relapse following curative therapy to primary tumors. It is thus essential to effectively assess the recurrence risk for personalized patient management. However, the current assessment capacity remains insufficient. We examined tumors produced in animals and samples derived from more than 1100 patients. A unique set of gene expression was observed following PC progression and a multigene panel consisting of 27 genes (Sig27gene) was constructed. Sig27gene is novel and robustly predicts PC relapse in two independent patient populations (*n* = 492 and *n* = 140) at *p* < 2 × 10^−16^. Sig27gene remains an independent risk factor of PC recurrence after adjusting for multiple clinical features. The novel and robust nature of Sig27gene support its translational potential to evaluate the risk of PC relapse in patients with primary PC.

**Abstract:**

IQGAP1 expression was analyzed in: (1) primary prostate cancer, (2) xenografts produced from LNCaP, DU145, and PC3 cells, (3) tumor of *PTEN^−/−^* and TRAMP mice, and (4) castration resistant PC (CRPC) produced by LNCaP xenografts and *PTEN^−/−^* mice. IQGAP1 downregulations occurred in CRPC and advanced PCs. The downregulations were associated with rapid PC recurrence in the TCGA PanCancer (*n* = 492, *p* = 0.01) and MSKCC (*n* = 140, *p* = 4 × 10^−6^) cohorts. Differentially expressed genes (*n* = 598) relative to IQGAP1 downregulation were identified with enrichment in chemotaxis, cytokine signaling, and others along with reductions in immune responses. A novel 27-gene signature (Sig27gene) was constructed from these DEGs through random division of the TCGA cohort into a Training and Testing population. The panel was validated using an independent MSKCC cohort. Sig27gene robustly predicts PC recurrence at (hazard ratio) HR 2.72 and *p* < 2 × 10^−16^ in two independent PC cohorts. The prediction remains significant after adjusting for multiple clinical features. The novel and robust nature of Sig27gene underlie its great translational potential as a prognostic biomarker to predict PC relapse risk in patients with primary PC.

## 1. Introduction

Prostate cancer (PC) is the top ranked male malignancy in the developed world [1]. The disease is developed from high grade prostatic intra-epithelial neoplasia (HGPIN) lesions that progress to PC and in some cases metastasis [2]. PCs are graded with Gleason score (GS) and World Health Organization (WHO) PC grading system (WHO grade group 1–5) or ISUP (the International Society of Urological Pathology) grade; WHO or its equivalent ISUP is GS-based [3,4,5]. Surgery is the primary curative therapy for localized PC; approximately 30% of patients will have PC relapse or biochemical recurrence (BCR) defined by an increase in serum prostate-specific antigen (PSA) [6]. A large percentage of BCR tumors will progress to metastatic PCs [7]. The standard of care for metastatic PCs is androgen deprivation therapy (ADT). Nonetheless, resistance in the form of castration-resistant PC (CRPC) commonly occurs [8,9]. A variety of regimens are available in managing CRPCs, including taxane-based chemotherapy, androgen receptor (AR)-targeting therapy involving either abiraterone or enzalutamide [9,10,11], and immunotherapy [12,13]. However, these treatments only produce modest survival benefits [9,14]. Conceptually, BCR remains the most desirable point of intervention before disease progression to metastasis and CRPC. Nonetheless, realization of this therapeutic option requires a better understanding of BCR.

Small G proteins are important oncogenic factors of PC. Cdc42 and Rac regulate cytoskeleton and reactive oxygen species, and activate MAP kinases and PI3K [15]. A critical mediator of Cdc42 and Rac is IQGAP1 which belongs to the IQ motif GTPase-activating scaffold proteins (IQGAPs). Both humans and mice have three IQGAP proteins, IQGAP1–3. Except for the WW motif, all domains among IQGAPs are highly conserved with reported homology ranging from 60% to 93% [16,17,18,19]. IQGAP1 stimulates ERK activation, associates with Cdc42 and Rac1, and stabilizes their GTP binding; IQGAP1 thus induces cytoskeleton dynamics [20], displays oncogenic activities and is upregulated in several cancers [21], including thyroid cancer [22], breast cancer [23], colorectal carcinoma [24], esophageal squamous cell carcinoma [25], hepatocellular carcinoma [26], and ovarian cancer [27]. While IQGAP2 shares an overall 62% homology with IQGAP1 with even higher levels of homology between their respective structural motifs except the WW domain [20,21], IQGAP2 surprisingly possesses tumor suppressive activities [19,21]. Nonetheless, IQGAP1 affects tumorigenesis in a complex manner. Its knockdown enhances T24 bladder cancer cells forming xenograft tumors in vivo and anchorage-independent cell growth in vitro [28]. Downregulation of IQGAP1 associates with high grade bladder cancer and poor patient survival [28]. Mechanistically, IQGAP1 downregulation increases TGFβ signaling in bladder cancer [28]. Similar mechanism is also used for IQGAP1 to suppress colorectal cancer liver metastasis via targeting tumor stroma [29]. Evidence thus supports tumor suppressive functions of IQGAP1.

In PC, IQGAP1 binds to p21 activated 6 (PAK6) in LNCaP cells [30] and reduces cell-cell adhesion in DU145 cells [31]. Additionally, upregulation of IQGAP1 was reported in metastasis produced by PC3 cells [32]. Nonetheless the clinical relevance of IQGAP1 in PC tumorigenesis remains unknown. We report here downregulation of IQGAP1 following PC tumorigenesis and progression in primary PC and CRPC. The downregulation is associated with BCR in both TCGA PanCancer PC and MSKCC cohorts. Furthermore, the IQGAP1 network is enriched with important oncogenic processes including reductions in immune responses. The network possesses capacity to robustly predict BCR, evident by the construction of Sig27gene.

## 2. Results

### 2.1. Downregulation of IQGAP1 following the Course of PC

To examine IQGAP1 expression in primary PCs, we obtained a set of primary PC tissues from Hamilton Health Sciences, consisting of advanced PCs (GS9–10, *n* = 14) and low-grade PCs (GS6–7, *n* = 13) (Appendix A). All PC pathologies were diagnosed by hospital pathologists and validated by a single pathologist. These tissues were among those collected by pathologists for our PC research [33,34]; however, the tissues used in this study were not included in our previous publications. IQGAP1 protein expression in these PCs was determined by IHC and quantified (Appendix A). In comparison to low grade PCs, significant reductions in IQGAP1 were observed in high grade PCs (Figure 1A,B).

To further determine IQGAP1 expression in PC, we took advantage of PC gene expression datasets available in the Oncomine^TM^ database (Compendia Bioscience, Ann Arbor, MI, USA). In both Liu [35] and Wallace [36] datasets, significant reductions of IQGAP1 mRNA in PC compared to normal prostate tissues were demonstrated (Figure 2A,B). Additionally, in the Grasso [37] and Chandran dataset [38], downregulations of IQGAP1 occurred in distant PC metastases compared to organ-confined PC (Figure 2C,D). Furthermore, IQGAP1 downregulation was also demonstrated using the Sawyers dataset [39] organized by R2: Genomics Analysis and Visualization Platform (http://r2.amc.nl
http://r2platform.com) (Figure 2E). Collectively, we demonstrate for the first time of IQGAP1 downregulation in PCs compared to normal prostate tissues and metastatic PCs (mPCs) compared to primary PCs.

The membrane expression of IQGAP1 in PCs was observed in PC xenografts produced from LNCaP, PC3, and DU145 cells (Figure 3A), primary PCs (Figure 3B), as well as tumors developed in TRAMP transgenic mice (Figure 3C). In comparison to PC3 cell-generated xenografts, those produced by LNCaP cells exhibit evidently more cell membrane IQGAP1 (Figure 3A). LNCaP [40] and PC3 [41] cells were derived from lymph node and bone metastases respectively; PC predominantly metastasizes to the bone [2]. While both LNCaP and PC3 cells can metastasize to the bone in mice via intracardiac injection, PC3 cells are widely regarded to be more aggressive. The apparent membrane expression of IQGAP1 in primary and xenograft PCs indicates a role of this proportion of IQGAP1 in PC pathology.

### 2.2. Association of IQGAP1 Downregulation with Therapy Resistance

Progression of ADT-resistance in the form of CRPC is a lethal progression of PC. It is thus interesting to see a significant reduction of IQGAP1 in patient-derived CRPCs compared to non-CRPCs in the Grasso dataset [37] within the Oncomine^TM^ database (Figure 4A). This reduction was also demonstrated at protein level in PC progressed in castrated *PTEN^−/−^* mice (CRPC) compared to PC produced in intact *PTEN^−/−^* mice (Figure 4B). The cell membrane location of IQGAP1 was clearly observed (Figure 4B). Furthermore, LNCaP cell-produced xenografts are androgen-sensitive, which will progress to CRPC following surgical castration in mouse [42]. In comparison to androgen-sensitive LNCaP tumors, LNCaP CRPCs are associated with a significant IQGAP1 downregulation at both the protein expression and mRNA expression level (Figure 4C–E). Taken together, this evidence supports a correlation of IQGAP1 reductions with the CRPC development.

Recurrence to the major curative therapy prostatectomy is the first major therapy resistance in PC progression. To investigate a potential association of IQGAP1 reduction with PC relapse following curative therapies, we downloaded IQGAP1 expression data along with PC biochemical recurrence (BCR) or progression information from the MSKCC and TCGA PanCancer Atlas PC datasets from cBioPortal. With optimal cutoff points defined by Maximally Selected Rank Statistics, PCs in the group with low IQGAP1 expression are associated with a rapid course of PC recurrence in both the TCGA (*p* = 0.001) and MSKCC (*p* = 4 × 10^−6^) cohorts (Figure 4F,G). IQGAP1 expression correlates with PC recurrence at hazard ratio (HR) 0.9996, 95% confidence interval (CI) 0.9996-1, and *p* = 0.0154 in the TCGA cohort and HR 0.2242, 95% CI 0.1006–0.4999, and *p* = 0.000258 in the MSKCC cohort. Collectively, we provide a comprehensive set of evidence for an association of IQGAP1 downregulation with therapy resistance in PC.

### 2.3. Enrichment of Oncogenic Pathways within the IQGAP1 Network

To characterize the association of IQGAP1 downregulation with PC progression, we have derived differentially expressed genes (DEGs) relative to IQGAP1 downregulation, following our established system [43,44]. In the TCGA PanCancer PC dataset, reduction of IQGAP1 mRNA expression at −1SD (standard deviation or z-score at −1) stratifies PCs into high or low risk group of PC recurrence with the high-risk group expressing reduced IQGAP1 (Appendix A). DEGs (*n* = 598) in the high-risk group (*n* = 72) vs. the low-risk group (*n* = 421) were derived at q < 0.001 and fold change ≥ |2| or log2Ratio ≥ |1| (Appendix A). IQGAP1 was expressed at log2Ratio = −0.94 and *q* = 8.24 × 10^−34^ in high-risk vs. low-risk PCs (Appendix A). 

We subsequently analyzed pathway enrichment in these DEGs using the Metascape network (https://metascape.org/gp/index.html#/main/step1) [45]. Top 20 non-redundant clusters are enriched, which include terms of GO (gene ontology) biological processes (BP) and KEGG pathways. The representatives of individual clusters (Figure 5A), the network of these enriched clusters (Figure 5B), and the details of enriched GO BP terms and KEGG pathways (Appendix A) are presented. The theme of enrichment centers on cytoskeleton dynamic-based processes (chemotaxis, cell adhesion, regulation of cell adhesion, extracellular matrix organization, responses to mechanical stimulus, cellular extravasation, cell morphogenesis, and cell-substrate adhesion), signaling responses (positive regulation of response to external stimulus, positive regulation of kinase activity, transmembrane receptor protein tyrosine kinase signaling pathway, and second-messenger-mediated signaling), and immune responses (regulation of cytokine production, leukocyte migration, and cytokine-mediated signaling pathway) (Figure 5A; please see Appendix A for details). The importance of these enriched clusters in tumorigenesis has been well established.

The above oncogenic contributions of the IQGAP1 network to PC is further supported by geneset enrichment analyses. Multiple immune reactions were downregulated in IQGAP1 DEGs, including interferon gamma (IFNγ) response, inflammatory response, IL2-STAT5 signaling, complement, TNFα, IFNα, IL6-JAK-STAT3 signaling, and TGFβ signaling (Figure 6, Table 1). The enhanced processes include Myc targets, DNA damage repair, and oxidative phosphorylation (Figure 6, Table 1). Collectively, we provide comprehensive evidence supporting the network associated with IQGAP1 downregulation in stimulating PC tumorigenesis and progression.

### 2.4. Construction of a Multigene Panel from the IQGAP1 Network in Predicting PC Recurrence following Prostatectomy

To further investigate the potential of the DEGs relative to IQGAP1 downregulation in association to PC, we have attempted to generate a signature or multigene panel from these DEGs to assess BCR. We randomly divided the TCGA PanCancer Atlas PC cohort into a Training (*n* = 344) and Testing (*n* = 148) population at the ratio of 7:3. The comparable demographics of both Training and Testing populations was demonstrated (Appendix A). Using the Training population, we carried out covariable selection among the 598 DEGs (Appendix A) for predicting PC recurrence (BCR) using Elastic-net within the R *glmnet* package. Twenty-seven genes (Sig27gene) were generated (Table 2). The considerations of using Elastic-net to control overfitting and others in variable selection have been outlined in Materials and Methods (see Section 4.11).

With the 27 genes being selected, we first examined Sig27gene in predicting BCR. Sig27gene scores for individual tumors were derived according to the formula: ∑(coef_i_ × Gene_iexp_)*_n_* (coef_i_: Cox coefficient of gene_i_, Gene_iexp_: expression of Gene_i_, *n* = 27). Coefs were obtained using the multivariate Cox model. The scores of Sig27gene robustly predict PC recurrence risk at HR 2.72, 95% CI 2.22–3.33, and *p* < 2 × 10^−16^ (Figure 7A). The score discriminates PC recurrence with time-dependent area under the curve (tAUC) values ranging from 88.5% at 10.8 months (88.5%/10.8 M) to 77.5%/47.7 months (Figure 7B). With the cutoff points determined using Maximally Selected Rank Statistics (Appendix A), Sig27gene efficiently stratifies patients in the Training cohort into groups with high- and low-risk for PC recurrence (Figure 7C).

### 2.5. Testing Sig27gene

Two strategies were employed to test Sig27gene. We first validated Sig27gene in the Testing population using Training-derived coefs. This signature score predicts PC recurrence risk in Testing cohort at HR 1.71, 95% CI 1.31–2.12, and *p* = 5.88 × 10^−5^ (Figure 7A); the best prediction efficiency is at tAUC value of 77.3%/47.8 M. In the full TCGA PanCancer PC population, Sig27gene with the coefs derived from the Training population evaluates PC recurrence at HR 2.23, 95% CI 1.92–2.60, and *p* < 2 × 10^−16^ (Figure 7A) with the associated tAUC values at 81.4%/11.5 M, 74.2%/21.9 M, 77.6%/31.7 M, and 77.1%/47.8 M. Sig27gene scores derived from Training effectively classify PCs with high risk of recurrence from those with low risk of recurrence in both the Testing and full TCGA cohort (Figure 7D,F).

We also used a second approach to reveal the full potential of Sig27gene in the prediction of PC relapse in the Testing and full TCGA cohorts via re-defining component gene coefs within each cohort using multivariate Cox analysis. These signature scores stratify high-risk PCs from low-risk PCs with robust efficiencies (Figure 7E,G). The prediction efficiencies were enhanced in both cohorts based on HR values (Figure 7A). The tAUC values are from 84.2%/13.1 M to 93%/47.8 M for Testing and 83%/11.5 M to 79%/31.7 M for full cohort (Figure 7B). Sig27gene can predict BCR risk in the Testing cohort as effective as its prediction of BCR in the Training cohort (comparing Figure 7E to Figure 7C). This enhancement supports the evaluations of Sig27gene biomarker potential via re-defining its signature scores, as it revealed that Sig27gene prediction ability is not diminished in a separate population, suggesting that Sig27gene was not majorly overfitted. Furthermore, the enhancement observed here indicates that Sig27gene parameters should be further defined with a much large cohort for its clinical assessment of BCR risk.

### 2.6. Validation of Sig27gene Using an Independent PC Cohort

The above two approaches were used to validate Sig27gene using an independent cohort, the MSKCC dataset. Using the Training coefs, Sig27gene scores significantly stratify PCs into a high- and low-risk group (Figure 8A) with the prediction efficiency at tAUC 73.2%/18.4 M (Figure 8C). The efficiency of Sig27gene scores in predicting PC relapse was robustly enhanced once the coefs for the component genes were re-derived from the MSKCC cohort (Figure 8B,C) with tAUC values ranging from 87.5% to 90.5% (Figure 8C), revealing the effectiveness of Sig27gene in estimating PC relapse in an independent PC cohort.

### 2.7. Evaluation of the Discriminative Performance of Sig27gene

We have used time-dependent ROC (receiver operating characteristic) curves to evaluate the discriminative ability of Sig27gene in Training, Testing, full cohort (Figure 7B), and MSKCC (Figure 8C) cohorts. tROC as a function of time serves the purpose to evaluate biomarker’s ability in discriminating the time to event survival including DFS [46]. Nonetheless, the proportion of patients with PC progressed (BCR) was 64 (18.6%) of 344 in training, 29/148 (19.6%) in testing, and 36/140 (25.6%) in the MSKCC cohort; these cohorts have a moderate degree of imbalance. It has been suggested that in extreme imbalanced datasets, in which the minority class is <1% of dataset, precision-recall (PR) curve is superior to ROC-AUC in evaluating marker’s discriminative performance [47]. A recent study has revealed that ROC-AUC correlates with PR-AUC regarding their performance with Pearson correlation coefficients of 0.88, 0.93, 0.96, 0.97, and 0.99 in datasets with a class prevalence of 0.091 (9.1%), 0.17, 0.25, 0.33, and 0.5 respectively [48]. Nonetheless, it was suggested to run both ROC-AUC and PR-AUC in imbalanced datasets particularly when imbalance is <5% [48]. To fully evaluate the discriminative abilities of Sig27gene in a separate (testing) and independent (MSKCC) cohort, we have constructed ROC and PR curves (Figure 9). Based on both ROC and PR curves, Sig27gene discriminates patients with BCR from those without disease progression with a solid efficiency, which is in line with the respective tROC curves (Figure 7B; Figure 8C). These analyses provide additional evidence for Sig27gene being not massively overfitted.

### 2.8. Sig27gene as an Independent Risk Factor of PC Recurrence

To examine the relationship of Sig27gene with clinical features in the estimation of PC relapse, multivariate Cox analysis was performed for Sig27gene, age at diagnosis, WHO prostate cancer grade (Grade I = GS6, Grade II = GS3 + 4; Grade III = GS4 + 3, Grade IV = GS8, and Grade V = GS9–10), margin status, and tumor stage. After adjusting for these clinical features, Sig27gene remain a strong risk factor of PC biochemical recurrence (Table 3).

To further explore the biomarker potential of Sig27gene, we were able to show that among its 27 component genes, 20 possess significant biomarker value in predicting PC recurrence (Table 4). Among the remaining seven component genes, both LAMP3 and KCNN3 are associated with PC relapse at *p* = 0.0616 and *p* = 0.0668, respectively. For the 20 significant component genes, they predict PC relapse with significant low *p* values up to 3.12 × 10^−8^ (Table 4). Furthermore, 10 of these 20 component genes are independent factors of PC biochemical recurrence after adjusting for age at diagnosis, WHO prostate cancer grade, margin status, and tumor stage (Table 4); this is impressive considering their individual status.

### 2.9. Characterization of Sig27gene

With the recent advances in the landscape of cancer-associated alterations in genome, methylation, and gene expression, cancers can be classified as integrative clusters (iClusters) [49]. PCs have been classified into iCluster 1, iCluster 2, and iCluster 3 [50]. PCs in iCluster 1 are enriched with ETV1 and ETV4 fusion, SHOP mutations, FOXA1 mutations, and CHD1 deletion, but lack ERG fusion [50]. iCluster 2 PCs are particularly enriched with ERG fusion and PTEN deletion [50]. iCluster 3 PCs contain ERG fusion [50]. TP53 hetero-deficiency and RB1 deletion are detected more frequently in iCluster 1 and iCluster 2 PCs [50]. In line with this knowledge, advanced PCs (GS ≥ 8) are much more frequent in iCluster 1 and iCluster 2 compared to iCluster 3 [50].

By using a recently established database GEPIA2 [51], we systemically determined the expression status of all component genes in PC in individual iClusters vs. matched normal prostate tissues. Among the 11 component genes with upregulations in relationship to IQGAP1 downregulation (Table 2), HAGHL, BIRC5, MXD3, PRR7, and RSG11 are significantly overexpressed in iCluster 1, iCluster 2, or both iClusters’ PCs compared to the matched prostate tissues (Figure 10). In comparison, among 16 downregulated genes relative to IQGAP1 under-expression (Table 2), FAM65B (RIPOR2), PI15, and HDAC9 are downregulated in either iCluster 1, iCluster 2, or both in comparison to the matched non-tumor tissues (Figure 10). The only exception is PCDHB8; while it is co-downregulated with IQGAP1 (Table 2), PCDHB8 is upregulated in iCluster 2 PCs compared to the matched normal controls (Figure 10). A point to take note is that none of the significant alterations occurred in iCluster 3 PCs (Figure 10). While the differences in IQGAP1 expression between PC and matched prostate tissues did not reach statistical significance (Appendix A), reductions of IQGAP1 in PC are apparent, particularly in iCluster 1 and 2 PCs (Appendix A), which provides additional support for downregulation of IQGAP1 in PCs. Collectively, the frequent alteration of Sig27gene component genes in iCluster 1 and iCluster 2 (Figure 10) supports the robustness of Sig27gene in predicting PC relapse.

We further determined component gene expressions in primary PCs without and with BCR using the Dunning dataset [52] organized by the R2: Genomics Analysis and Visualization Platform. Significant upregulations of HAGHL, VGF, RGS11, PPR7, and BIRC5 in primary PCs with BCR development were demonstrated (Figure 11). The directionality of these upregulations in BCR PCs compared to non-BCR PCs is consistent with their upregulations with IQGAP1 downregulation (Table 2). Apparent upregulations (*p* < 0.1) were observed for HIFX-AS1, MCTP1, MXD3, and PCDHB8 (Appendix A); apparent downregulation occurred in NOD2 (Appendix A).

We then examined all component gene expression in primary and metastatic PC (mPC) using the Sawyers dataset [39] within the R2: Genomics Analysis and Visualization Platform. In comparison to primary PCs, HAGHL, RGS11, PPR7, MXD3, BIRC5, ZFHX4, MCTP1, and PCDHB8 were upregulated (Figure 12), while RAB30, PI15, LAMP3, HDAC9, and KCNN3 were downregulated. Except for ZFHX4, MCTP1, and PCDHB8, the alterations of these genes are in line with their changes relative to IQGAP1 expression (Table 2). Apparent changes were also detected in DCST2, PLHNA4, and NOD2 (Appendix A).

We also analyzed the performance of Sig27gene in comparison to the most well-studied multigene panels, Oncotype DX and Prolaris (cell cycle progression/CCP). Both are commercially available to evaluate BCR risk at time of diagnosis [53,54,55,56,57] and following radical prostatectomy [58,59]. Oncotype DX consists of 12 testing and five reference genes [53]. Prolaris contains 31 testing and 15 reference genes [55]. Following our system, the panel scores for Oncotype DX using its 12 testing genes and Prolaris using all 31 testing genes were calculated; their abilities to stratify high- vs. low-risk PCs were analyzed with the optimized cutoff points. Both multigene panels effectively stratify PC relapse risk (Figure 13). In comparison, Sig27gene stratifies PC relapse risk using either the training score (Figure 7F) or optimized score (full cohort score, Figure 7G) more effectively than Oncotype DX and Prolaris (comparing the profiles, median months disease-free survival, and *p* values; Figure 7F,G and Figure 13). As Oncotype DX and Prolaris are real-time PCR-based quantification of gene expression, the analyses performed here may not fully reveal their potential. Nonetheless, the comparisons support Sig27gene as a robust multigene panel in prediction of PC relapse risk.

Taken together, the differential expressions of Sig27gene components in PCs vs. prostate tissues, non-BCR PCs vs. BCR-PCs, and primary PCs vs. mPCs support Sig27gene properties in predicting BCR. Additionally, the common upregulations of HAGHL, RGS11, PPR7, and BIRC5 in all three settings or comparisons (Figure 10, Figure 11 and Figure 12), suggesting their involvements in these processes. This possibility is intriguing considering both HAGHL and PPR7 being previously unknown for their oncogenic involvement in any cancer (Table 5).

### 2.10. Potential Oncogenic Functions of Sig27gene Component Genes

Among the 27 component genes of Sig27gene, eight genes have been reported in PC (Table 5), including four upregulated (VGF, RGS11, BIRC5 and LTC4S) and four downregulated (NOD2, PI15, LAMP3, and HDAC9) genes relative to IQGAP1 downregulation (Table 2; Table 5). VGF has been reported to facilitate radioresistance in DU145 and LNCaP cells [60] as well as resistance to tyrosine kinase inhibitors in lung cancer [61]. RGS11 is associated with TMPRSS2-ERG fusion [62], and is a biomarker of lung cancer [63]. BIRC5 or Survivin is a well-studied anti-apoptotic protein promoting PC and cancer metastasis [66,67]. LTC4S expression is upregulated in PC compared to normal tissues [68], and is a component gene in a immune signature associated with clinical response in breast cancer (Table 5) [69]. Among these four upregulated genes relative to IQGAP1 downregulation, VGF, RGS11, and BIRC5 were upregulated in primary PCs associated with BCR compared to non-BCR PCs (Figure 11), and the latter two were also upregulated in PC vs. normal prostate tissues (Figure 10) and mPCs vs. primary PCs (Figure 12). Collectively, the functionality of these upregulated component genes supports the concept of IQGAP1 downregulation in facilitating PC progression.

NOD2 facilitates innate immune response in prostate epithelial cells and likely plays a role in PC [74]; NOD2 was also implicated in immunosuppression of gastric cancer [75]. In line with this knowledge, NOD2 expression was evidently reduced in PCs compared to the matched normal prostate tissues (Appendix A) and distant metastasis compared to primary PC (Appendix A). Methylation of CpGs of the PI15 gene occurs in metastatic PC, which contributes to the stratification of metastatic PC from non-recurrent PCs [88]. Consistent with this report, PI15 expression was substantially reduced in primary PC vs. normal prostate tissues (Figure 10) and mPCs vs. primary PCs (Figure 12). Blood PI15 is a biomarker of cholangiocarcinoma (Table 5) [80]. LAMP3 was suggested to play a role in detoxification of cisplatin in CRPC [82] and associate with aggressive breast cancer (Table 5) [83]. In accordance with these reports, we detected a significant downregulation of LAMP3 in mPCs (Figure 12). Chromosome rearrangements in HDAC9 occur more frequently in high-risk PC compared to low-risk PCs [84]. Increases in HDAC9 were observed in basal bladder cancer [85]. Nonetheless, HDAC9 expression was significantly reduced in primary PC compared to normal prostate tissues (Figure 10) and mPCs compared to primary PCs (Figure 12). Taken together, with the exception of LAMP3, the genes that co-downregulated with IQGAP1 negatively impact PC, which reinforces a negative correlation of IQGAP1 with PC progression.

Nineteen Sig27gene component genes are unknown to participate in PC (Table 5). Nonetheless, 12 of these 19 genes are reported to function in tumorigenesis in general (Table 5), which include three upregulated and nine downregulated genes (Table 2; Table 5). LINC01089, MXD3, and H1FX-AS1 are upregulated component genes (Table 2). Evidence supports MXD3 in promotion of medulloblastoma [65] and both LINC01089 [64] and H1FX-AS1 [70] display tumor suppressive functions (Table 5). ZFHX4 is one of the 9 under-expressed genes and is a susceptibility locus of cutaneous basal cell carcinoma (Table 5) [81]. Both RRAGC [78] and TFEC regulate mTOR activation with the latter affects mTOR via lysosome biogenesis (Table 5) [79]. With LAMP3 also functioning in lysosome, Sig27gene likely affects lysosome biology and mTOR activation. Evidence supports negative impacts on tumorigenesis for the rest of six downregulated genes (Table 5), including FPR3 [71], RAB30 [72], RIPOR2 (FAM65B) [73], PLXNA4 [76,77], MCTP1 [86], and KCNN3 [87]. Collectively, the positive and negative impacts of these 12 genes unknown to PC on tumorigenesis are generally in line with the notion of IQGAP1 negatively associating with PC.

In addition to Sig27gene affecting mTOR and lysosome processes as discussed above, the signature also affect immune reactions, particularly innate immune response. NOD2 facilitates innate immune response in prostate epithelial cells [74], and is likely downregulated in PC (Appendix A), and co-reduced in PC with IQGAP1 (Table 2). PLXNA4 inhibits tumor cell migration, induces innate immune responses via working with Toll-like receptor (Table 5) [76,77], and is also co-downregulated with IQGAP1 in PC (Table 2).

## 3. Discussion

Biochemical recurrence (BCR) remains a critical issue in PC management; this is not only due to this progression being the initial point of therapy resistance leading to poor prognosis but also because this is conceptually the most effective point of intervention. While mechanisms underlying BCR have been extensively investigated, with numerous biomarkers and systems in place to assess BCR [89] including miRNAs [90], the current capacity in predicting BCR is clearly not sufficient. Currently, there are two commercially available multigene panels (Oncotype DX and Prolaris) to assess PC relapse risk at time of diagnosis or after radical prostatectomy (RP). Although both are not in routine clinical applications, evidence supports both improving the risk assessment [53,54,55,56,57,58,59]. In comparison to both panels, Sig27gene can stratify PCs with high-risk recurrence from those with low risk as effective as, if not superior to, Oncotype DX and Prolaris in the risk stratification (comparing Figure 7F,G to Figure 13). It is thus tempting to suggest that Sig27gene can be used to evaluate BCR risk following RP; this knowledge will enhance patient management. An appealing scenario could be the combined use of Sig27gene, Oncotype DX, and Prolaris to predict the risk of PC relapse, as all three panels estimate the risk based on different aspects of PC. Oncotype DX focuses on stroma, cellular organization, and androgen signaling [53]; Prolaris was constructed via modeling cell proliferation [55]; and Sig27gene includes immune pathways (mTOR and innate immunity, see later discussions). Additionally, BIRC5 is the only common component gene in Sig27gene and Prolaris; Sig27gene is thus likely a valuable addition to Oncotype DX and Prolaris.

This research represents a novel attempt in improving BCR risk assessment through the angle of IQGAP1. We have approached this research because the dynamics of cytoskeleton organization is essential for tumor progression through processes of epithelial mesenchymal transition (EMT) and mesenchymal epithelial transition (MET) as well as communications with microenvironment [91]. IQGAP1 plays an important role in cytoskeleton reorganization via stabilization of the GTP-bound Cdc42 and Rac1 [20]. Most published evidence supports IQGAP1 in promoting tumorigenesis [21], a concept that is in agreement with the limited number (*n* = 4 in PubMed) of PC studies. However, we provide comprehensive evidence suggesting a potential tumor suppressive role of IQGAP1 in PC, which includes (1) downregulation of IQGAP1 in PCs vs. prostate tissues, mPC vs. primary PCs, and CRPC vs. androgen-sensitive PCs; (2) the association of IQGAP1 reduction with PC biochemical recurrence in two independent cohorts, MSKCC and TCGA PanCancer; and (3) the enrichment of pathways or processes underlined by cytoskeleton dynamics (Figure 5, Table 1). The mechanisms underlying IQGAP1 downregulation are likely complex. It is possible that IQGAP1 reduction is a result of PC tumorigenesis and development.

The mechanisms responsible for IQGAP1 downregulation-affected cytoskeleton dynamics remain unclear. It is unlikely that these actions are mediated through Cdc42 and Rac1, as this connection would favor oncogenesis. While these mechanisms require further investigations, it is tempting to propose that (1) IQGAP1 regulates PC cell adhesion or the related processes indirectly via its DEGs or network and (2) the cell surface IQGAP1 is attributable to these actions. The second possibility is appealing as the location is relevant to cell adhesion and IQGAP2, a PC suppressor [19], was largely detected on PC cell surface [19]. The high level of homology with IQGAP2 supports the membrane location of IQGAP1 and this proportion of IQGAP1 in suppression of PC. This concept is supported by IQGAP1 being more abundantly localized to xenograft PC cell membrane produced by LNCaP cell compared to those generated by PC3 cells, a more aggressive PC cell line. Additionally, upregulations of IQGAP1 in metastasis of PC3 cell-generated tumors, which was previously reported [32], was largely intracellular IQGAP1. Intriguingly, pro-tumorigenic roles of IQGAP1 were observed in breast cancer [23]; its cytosolic and nuclear expressions, where IQGAP1 was co-localized with BRCA1, were detected in triple negative breast cancer [92]. IQGAP1 promoted thyroid cancer and was largely expressed in the cytosol [22]. Similarly, the cytosolic expression of IQGAP1 in colorectal cancer was associated with its pro-oncogenic functions [24]. In non-small cell lung cancer, the cytoplasmic and nuclear expressions of IQGAP1 were correlated with lymph node metastasis and poor overall survival [93]. Evidence thus suggests that different cellular expressions of IQGAP1 might in part explain the PC-facilitative function of IQGAP1 reported by others and the PC-suppressive roles of IQGAP1 observed here. While IQGAP1 may facilitate PC, our study supports its inhibitive role towards PC, a concept that is in accordance with molecular events affected by IQGAP1 downregulation.

The major pathways or processes affected by the IQGAP1 network include the reductions of immune signaling pathways (Figure 6, Appendix A). Appealingly, these reductions in immune signaling within the DEGs are also reflected in the signature constructed. It is intriguing to see that both NOD2 and PLXNA4 are among the downregulated component genes in Sig27gene and both induce innate immune reactions (Table 5) [74,77]. Modulation of immune profiles to set up permissive microenvironment is critical for PC initiation and progression [94,95], which might be relevant to the cell membrane expression of IQGAP1.

Another novel feature of Sig27gene is the modulation of mTOR activation and regulation of lysosome biology. Lysosome is well-regarded to induce mTOR activation in response to nutrient cues [96]. Collectively, modulation of multiple critical oncogenic processes is likely a major attributor to the robust efficiency of Sig27gene in predicting PC recurrence. An intriguing feature of Sig27gene is the clusters of 3 component genes at 5q31.3 and 3 (PRR7, MXD3, and LTC4S) at 5q35.3, and 2 (PI15 and ZFHX4) at 8q21.13 (Table 2). The importance of these clusters remains unclear.

Finally, in addition to the novelties described above, Sig27gene is composed of a large proportion of component genes (*n* = 19) novel to PC and tumorigenesis in general (*n* = 7) (Table 5). These genes are likely relevant at least to PC. For instance, HAGHL (Hydroxyacylglutathione Hydrolase Like), LCN12 (Lipocalin 12), DCST2 (DC-STAMP Domain Containing 2), and PRR7 (Proline Rich 7, Synaptic) not only significantly predicts PC recurrence but also remain risk factor status after adjusting for age at diagnosis, WHO prostate cancer grade, margin status, and tumor stage (Table 4). Both HAGHL and PRR7 were upregulated in PC compared to normal prostate tissues, PCs at risk of BCR development compared to those with low BCR risk, and mPC compared to primary PCs (Figure 10, Figure 11 and Figure 12). Both MXD3 and PCDH8 are novel in PC; they were upregulated in PCs compared to prostate tissues (Figure 10) and mPCs compared to primary PCs (Figure 12). Their functions in PC indeed warrant future investigations.

## 4. Materials and Methods

### 4.1. Collection of PC Tissues

PC tissues were obtained from Hamilton Health Sciences, Hamilton, ON, Canada under approval from the local Research Ethics Board (REB# 11-3472).

### 4.2. Cell Culture

LNCaP, PC3, and DU145 cells were purchased from American Type Culture Collection (ATCC, Manassas, VA, USA) and cultured in RPMI1640, F12 or MEM respectively, followed with supplementation of 10% FBS (Sigma Aldrich, Oakville, ON, Canada) and 1% penicillin-streptomycin (Thermo Fisher Scientific, Burlington, ON, Canada). The cell lines were authenticated (Cell Line Authentication Service, ATCC), and routinely checked for mycoplasma contamination (a PCR kit from Abcam, Toronto, ON, Canada, Cat#: G238).

### 4.3. Formation of Xenograft Tumors

Xenografts were generated as previously described [97,98,99,100]. Briefly, LNCaP, DU145 or PC3 cells (3 × 10^6^) in 0.1 mL culture media were mixed with Matrigel mixture (BD) at 1:1 (volume: volume), and implanted subcutaneously (s.c.) into the flank of NOD/SCID mice (6-weeks old males with five mice per group; The Jackson Laboratory, Bar Harbor, ME, USA). Tumor growth was monitored. Tumor size was weekly measured using calipers and calculated as V = L × W^2^ × 0.52. Endpoints were defined as tumor volume ≥ 1000 mm^3^. Mice were euthanized by CO_2_ followed by cervical dislocation. All animal experiments were carried out based on the protocols approved by the McMaster University Animal Research Ethics Board (AUP#: 16-06-24).

### 4.4. Generation of CRPC in Animal Models

LNCaP cells (5 × 10^6^)-derived s.c. xenografts were generated in NOD/SCID mice (The Jackson Laboratory) with tumor volume determined [34]. Tumor growth was measured by serum PSA levels (PSA kit, Abcam, Toronto, Ontario, Canada). Surgical castration was performed when tumor reached 100–200 mm^3^. Serum PSA was determined before and following castration. Rise in serum PSA indicates CRPC growth.

Prostate-specific *PTEN^−/−^* mice were generated by crossing *PTEN^loxp/loxp^* (C;129S4-*Pten^tm1Hwu^*/J; the Jackson Laboratory) mice with PB-Cre4 mice (B6.Cg-Tg(Pbsn-cre)4Prb, the NCI Mouse Repository) following our published conditions [101]. Surgical castration was performed when mice were 23 weeks old and subsequently monitored for 13 weeks. All animal protocols were approved by the McMaster University Animal Research Ethics Board.

The male TRAMP animals (C57BL/6-Tg (TRAMP)8247 Ng/J; the Jackson Laboratory) and the nontransgenic littermates were routinely obtained as [TRAMP C57BL/6 × C57BL/6] F_1_. Ear clips were taken and incubated in digest buffer (1 M Tris-HCl pH 8.0, 0.5 M EDTA pH 8.0, 3 M NaCl, 10% SDS, 10 mg/mL proteinase K) overnight at 55 °C. Mouse DNA was then extracted with ethanol precipitation. PCR was performed with DreamTaq Hot Start PCR Mastermix (Invitrogen, Burlington, Ontario, Canada) following PCR condition established by the Jackson Laboratory. Amplified PCR products were run on 2.25% agarose gel and visualized with ultraviolet transilluminator. Once genotype is confirmed, male TRAMP mice is maintained and monitored for 34 weeks or when endpoints were reached (weight loss, palpable tumor or apparent physical distress). All animal protocols were approved by the McMaster University Animal Research Ethics Board.

### 4.5. Immunohistochemistry (IHC)

Xenograft tumor tissues were prepared using formalin-fixed paraffin-embedded (FFPE) procedure [100,102]. Slides were cut from FFPE tissue blocks, deparaffinized in xylene, and cleared in an ethanol series. Antigens were retrieved through heat treatment in sodium citrate buffer (pH = 6.0), followed by blocking in PBS containing 1% BSA and 10% normal goat serum (Vector Laboratories, Burlington, Ontario, Canada) for 1 h and incubation with anti-IQGAP1 antibody (1:800, Cell Signalling, Danvers, MA, USA) overnight at 4 °C. Secondary antibody (biotinylated goat anti-rabbit IgG) and Vector ABC reagent (Vector Laboratories) were then applied. Secondary antibody alone was used as negative control. Chromogenic reaction was developed with diaminobenzidine (Vector Laboratories); slides were counterstained with hematoxylin (Sigma Aldrich). Images were analyzed using ImageScope software (Leica Microsystems Inc.); staining intensity was quantified as HScores using the formula [H-Score = (% Positive) × (intensity) + 1] [100,102].

### 4.6. Analysis of IQGAP1 mRNA Expression

The PC datasets were retrieved from the Oncomine^TM^ database (https://www.oncomine.org/). IQGAP1 mRNA expression data was analyzed in PC vs. prostate tissues, metastasis vs. local PC, and CRPCs vs. non-CRPC tumors. IQGAP1 expression was also evaluated using R2: Genomics Analysis and Visualization Platform (http://r2.amc.nl
http://r2platform.com).

Total RNA was isolated from xenografts tissues from LNCaP intact or LNCaP CRPC animals with the Iso-RNA Lysis Reagent (5 PRIME); reverse transcription was performed using Superscript III (Thermo Fisher Scientific). Quantitative real-time PCR was performed using the ABI 7500 Fast Real-Time PCR System (Applied Biosystems, Foster, California, USA) using SYBR-green (Thermo Fisher Scientific) with the following primers: IQGAP1 Forward 5′-AAGAAGGCATATCAAGATCGG-3′, and reverse 5′- CCTCAGCATTGATGAGAGT-3′; β-Actin forward 5′-TGAAGGTGACAGCAGTCGGT-3′, and reverse 5′-TAGAGAGAAGTGGGGTGGCT-3′. Fold changes were calculated using the formula: 2^−ΔΔCt^.

### 4.7. cBioPortal Database

The cBioPortal [103,104] (http://www.cbioportal.org/index.do) database contains the most well-organized cancer genetics for various cancer types. The TCGA PanCancer Atlas PC dataset contains *n* = 492 tumors. Tumors have been removed by prostatectomy with RNA expression profiled by RNA sequencing (RNA-seq). The suitability of the dataset for PC recurrence-related biomarker studies has been demonstrated [105]. The MSKCC [39] dataset was also used. 

### 4.8. Pathway Enrichment Analysis

Enrichment analyses were carried out using Metascape [45] (https://metascape.org/gp/index.html#/main/step1); geneset enrichment was performed using fgsea in Galaxy (https://usegalaxy.org/).

### 4.9. Cutoff Point Estimation

Cutoff points to stratify patients into a high- and low-risk group were estimated by Maximally Selected Rank Statistics (the *Maxstat* package) in R.

### 4.10. Regression Analysis

Cox proportional hazards (Cox PH) regression analyses were carried out with the R survival package. The PH assumption was tested.

### 4.11. Establishing of a Multigene Panel Predicting PC Biochemical Recurrence

IQGAP1-associated differential expressed genes (DEGs, *n* = 598) were derived from the TCGA PanCancer Atlas PC dataset within the cBioPortal database [103,104] (https://www.cbioportal.org/). The dataset was randomly divided into a Training and Testing population at the ratio of 7:3 using R. DEGs were selected for best prediction of BCR using Elastic-net logistic regression (the glmnet package in R), which was based on a few factors. (1) The high-dimensional nature (DEGs, *n* = 598) of our data requires control overfitting. This issue is statistically managed through shrinkage or regularization: L1 and L2 regularization. Ridge regression shrinks regression coefficients by penalizing the sum of the squared coefficients (L2 regularization), while LASSO (Least Absolute Shrinkage and Selection Operator) shrinks regression coefficients via penalizing the sum of the absolute coefficients (L1 regularization). Elastic-net uses both L1 and L2 regularization to shrink regression coefficients. (2) The number of variable (DEGs, *n* = 598) exceeds the number of patients in the TCGA PanCancer cohort (*n* = 492); Elastic-net is superior to LASSO in performing variable selection in this dataset. (3) Ridge regression is without capacity for variable selection and LASSO selects one covariate among a group of related variables; this will reduce the signature’s biomarker potential. Elastic-net can select correlated covariates (see the *glmnet* package and “The Elements of Statistical Learning” by Hastie et al.: https://web.stanford.edu/~hastie/Papers/ESLII.pdf). The mixing parameter of α in Elastic-net can be set at α = 0, at which it runs as Ridge regression and when α = 1, Elastic-net operates as Lasso. We thus set α = 0.5. Because of variation in variable selection during individual rounds of selection, we performed six rounds of selection and all unique genes obtained were combined into the final multigene panel Sig27gene.

### 4.12. Assignment of Signature Scores to Individual PCs

Component genes (*n* = 27) of Sig27gene were examined for associations with BCR using multivariate Cox PH regression with the R Survival package. The signature scores for individual tumors were given using the formula: Sum (coef_1 ×_ Gene_1exp_ + coef_2_ × Gene_2exp_ + … …+ coef_n_ × Gene_nexp_), where coef_1_ … coef_n_ are the coefs of individual genes and Gene_1exp_ … … Gene_nexp_ are the expression of individual genes.

### 4.13. Examination of Gene Expression

The expression of Sig27gene component genes in PC and normal prostate tissues was determined using a newly established GEPIA2 dataset [51]. Their expression in PCs with and without BCR as well as in local vs. distant metastatic PCs were also determined using R2: Genomics Analysis and Visualization Platform (http://r2.amc.nl
http://r2platform.com).

### 4.14. Statistical Analysis

Kaplan-Meier survival analyses and logrank test were carried out using the R Survival package and with tools provided by cBioPortal. Univariate and multivariate Cox regression analyses were run with the R survival package. Time-dependent receiver operating characteristic (tROC) analyses were performed using the R *timeROC* package. ROC and precision-recall (PR) curves were produced using the PRROC package in R. Gaussian distribution of IHC data was tested using Shapiro-Wilk test, D’Agostino-Pearson normality test and Anderson-Darling test. Non-parametric two-tailed Mann-Whitney test, two-tailed Student’s *t*-test, and one-way ANOVA were also used. A value of *p* < 0.05 is considered statistically significant.

## 5. Conclusions

Based on our best knowledge, this is the first analysis IQGAP1 expression in primary PCs and CRPCs produced in vivo. We provide the first demonstration for IQGAP1 downregulations in PC compared to normal prostate tissues, high-grade PCs compared to low-grade PCs, metastatic PCs compared to primary PCs, and CRPCs compared to hormone-naïve PCs. Furthermore, IQGAP1 downregulations are associated with reductions in disease-free survival or PC relapse in two independent cohorts (*n* = 492 and *n* = 140). Decreases in IQGAP1 expression are associated with network alterations consisting of *n* = 598 DEGs that affect pathways important to PC progression. These DEGs contain a 27-gene panel (Sig27gene) which robustly predicts PC relapse in two independent cohorts (*n* = 492 and *n* = 140) at *p* < 2 × 10^−16^. The prediction is independent of WHO PC grades, tumor stage, surgical margin, and age at diagnosis. Sig27gene not only is a novel multigene panel but also predicts PC relapse with a high level of certainty. The novelty of Sig27gene is also attributed to its component genes; 19 of the 27 component genes are novel to PC. Among these novel PC genes, HAGHL, MXD3, and PRR7 are upregulated relative to IQGAP1 downregulation; all three genes are upregulated in primary PC compared to normal prostate tissues and metastatic PCs compared to primary PC; and HAGHL and PRR7 are also over-expressed in PC with high relapse risk compared to those with low relapse risk. Within the 19 genes novel to PC, RIPOR2, RAB30, and KCNN3 are co-downregulated with IQGAP1, while RIPOR2 is under-expressed in primary PCs compared to normal prostate tissues, RAB30 and KCNN3 are downregulated in mPCs compared to primary PCs. Collectively, this research produces three novelties: (1) IQGAP1 downregulations following PC progression, (2) a novel and robust multigene panel (Sig27gene) in assessing PC relapse, and (3) 6 novel candidates of PC genes. This work may have a profound impact on PC research and patient management.

## 6. Patents

This research has resulted in a USA provisional patent application.

## Figures and Tables

**Figure 1 cancers-13-00430-f001:**
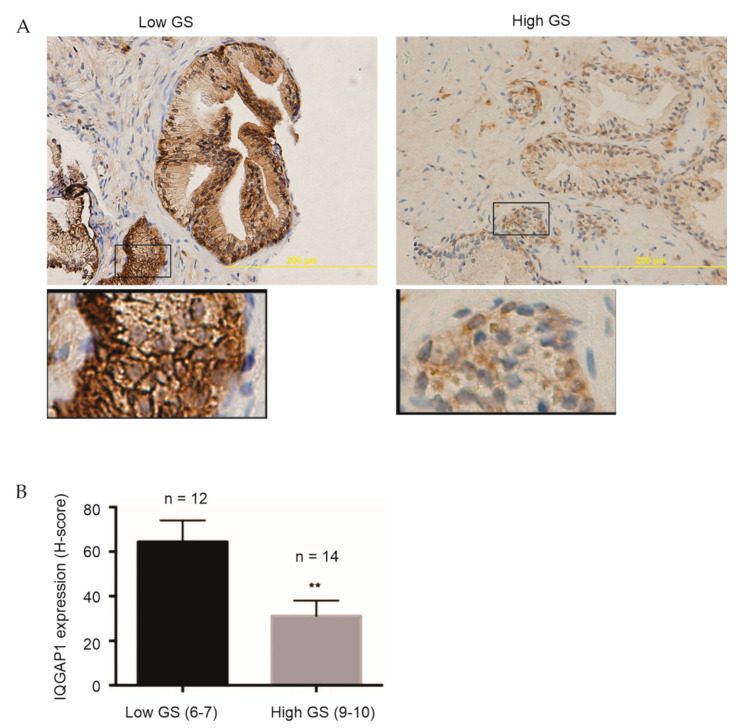
Downregulation of IQGAP1 in advanced PCs. Primary PCs with low Gleason scores 6–7 (GS 6–7) and high GS 9–10 (see Appendix A for details) were stained for IQGAP1 by IHC. Typical images (**A**) and quantification (**B**), means ± SD (standard deviation), are graphed. **: *p* < 0.01 by 2-tailed Mann-Whitney test. Images were analyzed using Aperio ImageScope software (Leica Microsystems Inc., Concord, Ontario, Canada); staining intensity was quantified as Histo-scores (HScores). Stromal regions (control) were used as baseline when analyzing final H-score for each sample.

**Figure 2 cancers-13-00430-f002:**
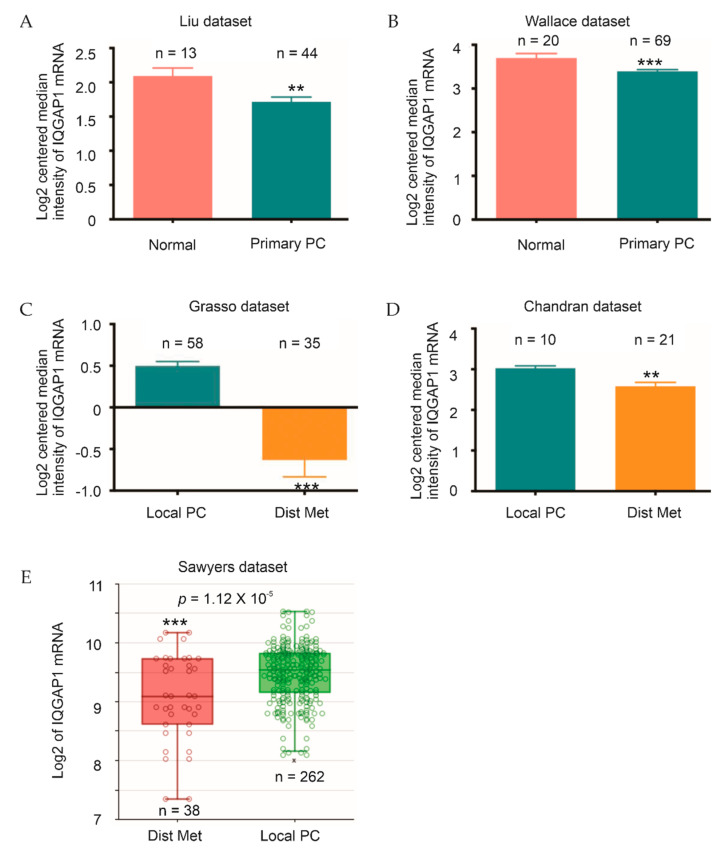
Decreases in IQGAP1 mRNA expression following the course of PC. (**A**,**B**) The Liu and Wallace datasets of PC microarray studies within Oncomine^TM^ were analyzed for IQGAP1 expression in primary PC and normal prostate tissues. **: *p* < 0.01 and ***: *p* < 0.001. (**C**,**D**) The indicated microarray datasets of PC from Oncomine^TM^ were analyzed for IQGAP1 expression in primary PCs and distant metastasis PCs. (**E**) IQGAP1 expression in metastatic PCs and primary PCs in the Sawyers dataset organized by the R2: Genomics Analysis and Visualization Platform. **: *p* < 0.01 and ***: *p* < 0.001 by 2-tailed Student’s *t*-test.

**Figure 3 cancers-13-00430-f003:**
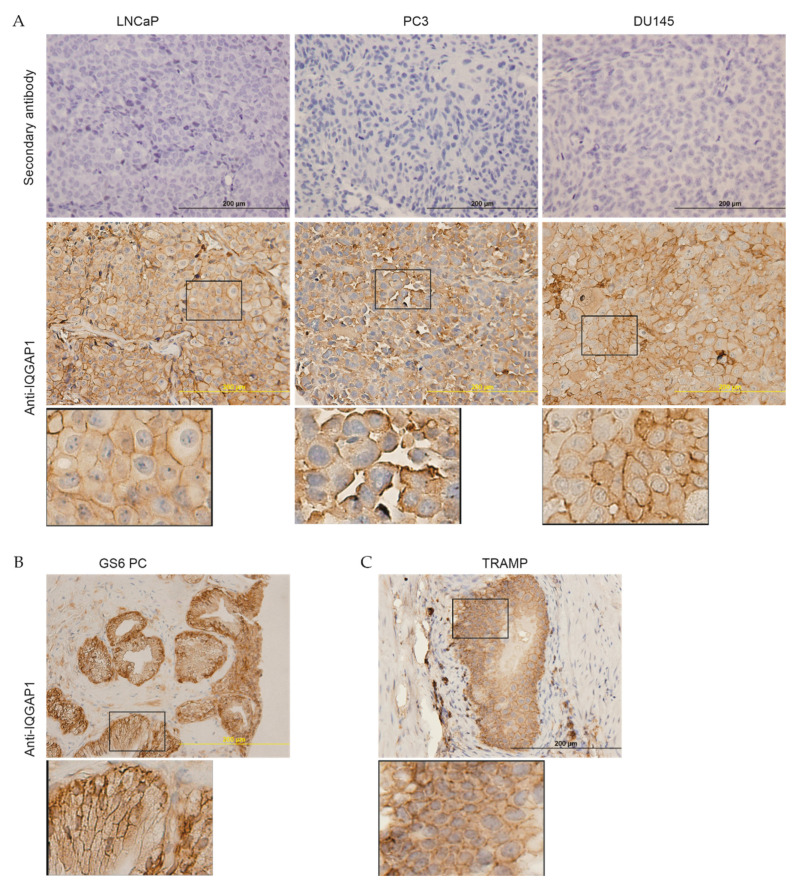
Membrane expression of IQGAP1 in prostate tumors. (**A**) Xenografts were produced from LNCaP (*n* = 3), PC3 (*n* = 5), and DU145 (*n* = 5) PC cells in NOD/SCID mice, followed by IHC examination for IQGAP1. Secondary antibody was used as a negative control. Typical images for the indicated xenografts are presented. (**B**) IHC staining of IQGAP1 in low GS (GS 6–7) primary PCs. (**C**) Tumors produced in TRAMP transgenic mice (*n* = 5). The marked regions are enlarged 3-fold and placed underneath the individual panels. IHC images were analyzed using Aperio ImageScope software (Leica Microsystems Inc.); staining intensity was quantified as Histo-scores (HScores). Stromal regions (control) were used as baseline when analyzing final H-score for each sample.

**Figure 4 cancers-13-00430-f004:**
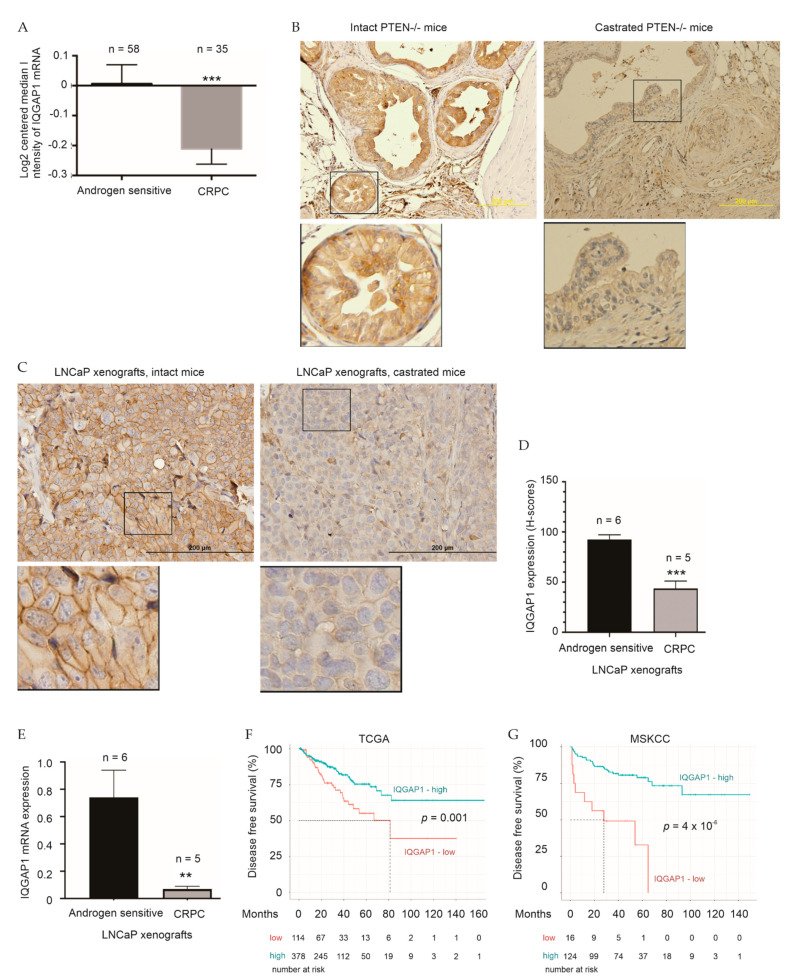
Downregulation of IQGAP1 associates with therapy resistance of PC. (**A**) IQGAP1 mRNA expression in the Grasso dataset (Oncomine^TM^) in androgen sensitive PCs and CRPCs was analyzed. ***: *p* < 0.001 by 2-tailed Student’s *t*-test. (**B**) Prostate specific *PTEN^−/−^* mice at 23 weeks old were either remained as intact status or castrated and maintained for additional 13 weeks. PCs from intact mice (*n* = 2) and CRPCs from castrated mice (*n* = 2) were stained for IQGAP1 using IHC. Typical images along with 3-folds enlargement of the indicated regions are included. (**C**,**D**) LNCaP xenografts were generated in NOD/SCID mice. Mice were either untreated or castrated when tumors at 100–200 mm^3^, followed by monitor of PSA increases. Typical images with 3-folds enlargement of the indicated regions (**C**) and quantifications (**D**) are shown. Shapiro-Wilk test was used to confirm Gaussian distribution of data. ***: *p* < 0.001 by 2-tailed Student t-test. IHC images were analyzed using Aperio ImageScope software (Leica Microsystems Inc.); staining intensity was quantified as Histo-scores (HScores). Stromal regions (control) were used as baseline when analyzing final H-score for each sample. (**E**) IQGAP1 mRNA expression in LNCaP intact vs. CRPC tumors were analyzed using RT-PCR. **: *p* < 0.01. (**F**,**G**) IQGAP1 mRNA expression data, determined by RNA-seq, was retrieved along with the relevant clinical data from the TCGA PanCancer Atlas and MSKCC dataset within cBioPortal. Cutoff points to separate the individual cohort into a high and low recurrence risk group were defined by Maximally Selected Rank Statistics using the R *Maxstat* package. Kaplan Meier curves and log-rank test were performed using the R Survival package (CRAN - Package survival (r-project.org)). Numbers at risk are indicated.

**Figure 5 cancers-13-00430-f005:**
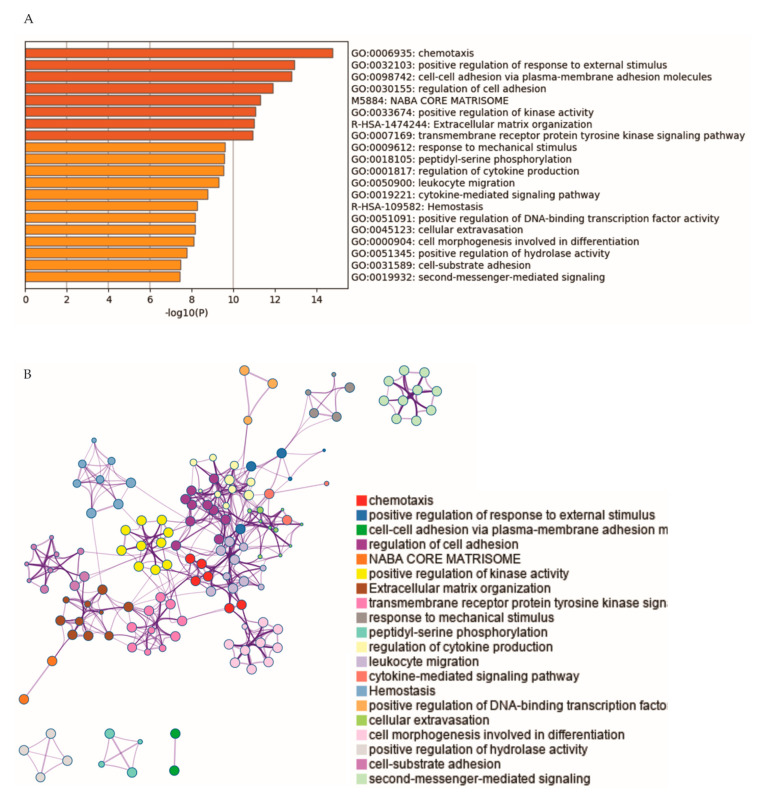
Pathway enrichment of IQGAP1 DEGs. (**A**) Representatives of top 20 enriched clusters of GO, biological process terms and KEGG pathways are shown. (**B**) Network relationship of those enriched clusters. Analyses were carried out with Metascape [45].

**Figure 6 cancers-13-00430-f006:**
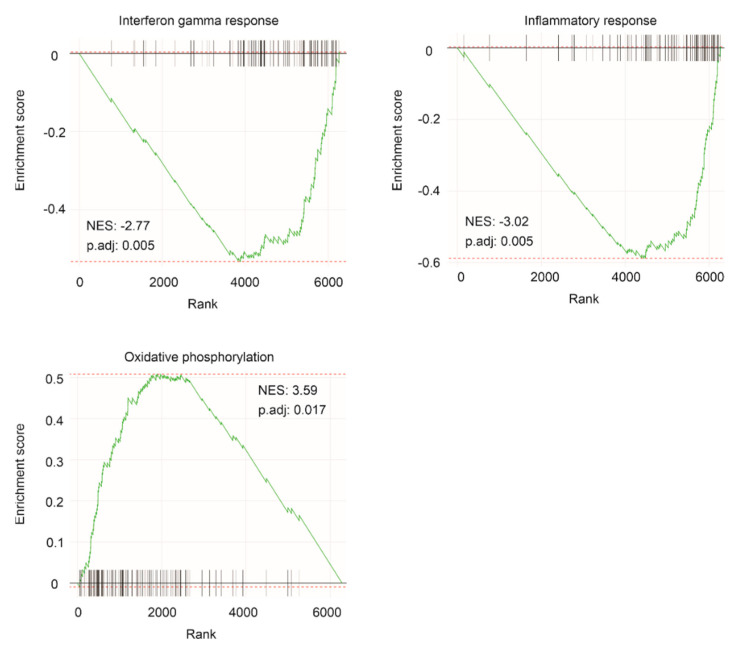
Geneset enrichment. IQGAP1 DEGs relative to IQGAP1 downregulation were defined at *q* < 0.0001 and analyzed for geneset enrichment among human hallmark gene set. Three enriched genesets functioning in interferon gamma response, inflammatory response and oxidative phosphorylation are included.

**Figure 7 cancers-13-00430-f007:**
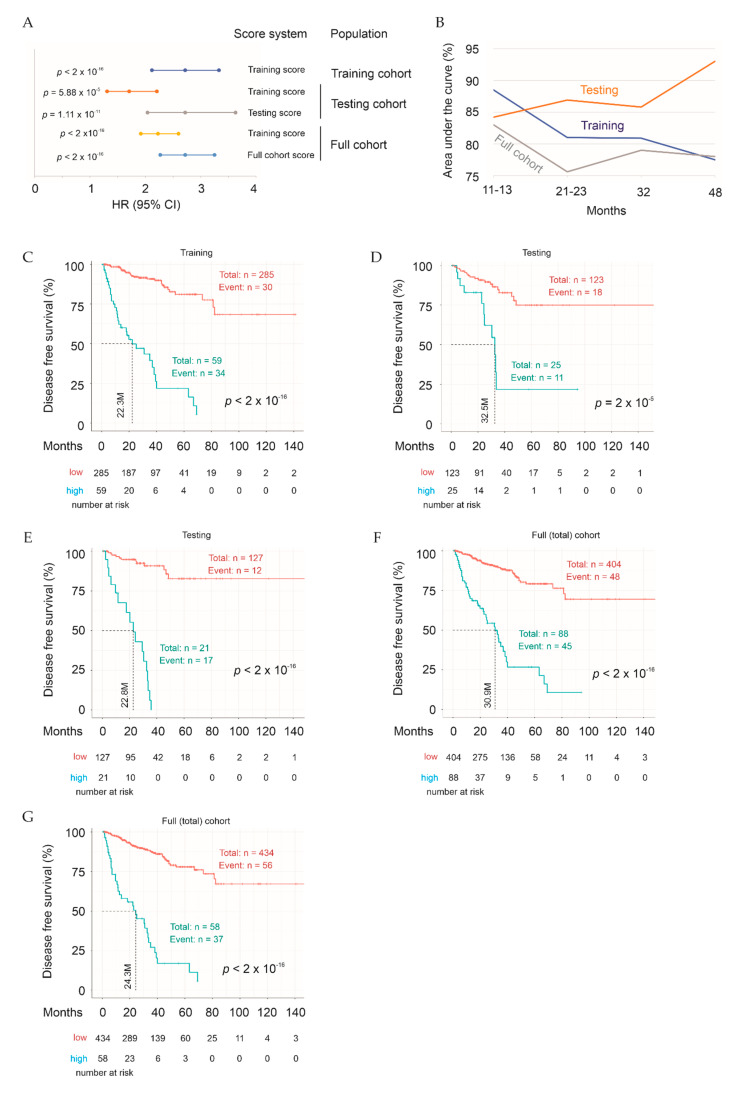
Sig27gene robustly stratifies the risk of PC recurrence. (**A**) HR, 95% CI, and *p* values for prediction of PC biochemical recurrence in the indicated populations are shown. Signature scores were either from the Training group (“Training score”), Testing cohort (“Testing score”), or full TCGA cohort (“Full cohort score”). (**B**) Time-dependent ROC (receiver operating characteristic) curve; time-dependent area under the curve (AUC) values for the indicated cohorts are shown. (**C**) Kaplan Meier curve for Training cohort was produced based on the cutoff point of Sig27gene determined by Maximally Selected Rank Statistics. Statistical analyses were performed using logrank test. (**D**,**E**) Kaplan Meier curves for the Testing cohort were produced using the Training scores (**D**) or Testing scores (**E)** of Sig27gene. (**F**,**G**) Examination of Sig27gene in the full TCGA PanCancer PC dataset using the Training scores (**F**) and full cohort scores (**G**). All Kaplan Meier analyses and logrank tests were performed using the R Survival package.

**Figure 8 cancers-13-00430-f008:**
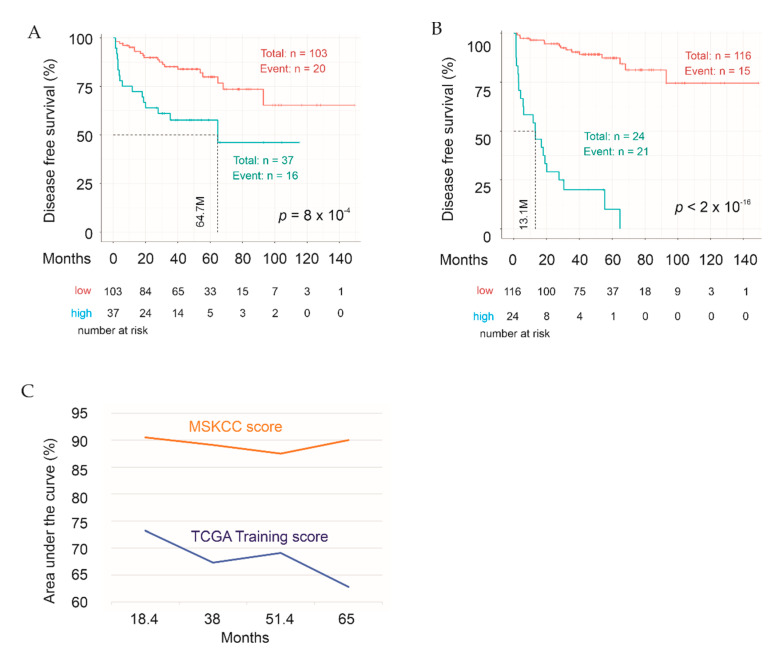
Validation of Sig27gene with the independent MSKCC PC cohort. (**A**,**B**) Sig27gene was analyzed for the stratification of PC recurrence risk in MSKCC dataset using scores defined either from the TCGA Training cohort (TCGA Training score) (**A**) or from the MSKCC cohort (MSKCC score) (**B**). (**C**) Time-dependent AUC for the indicated score systems in prediction of PC recurrence in MSKCC cohort.

**Figure 9 cancers-13-00430-f009:**
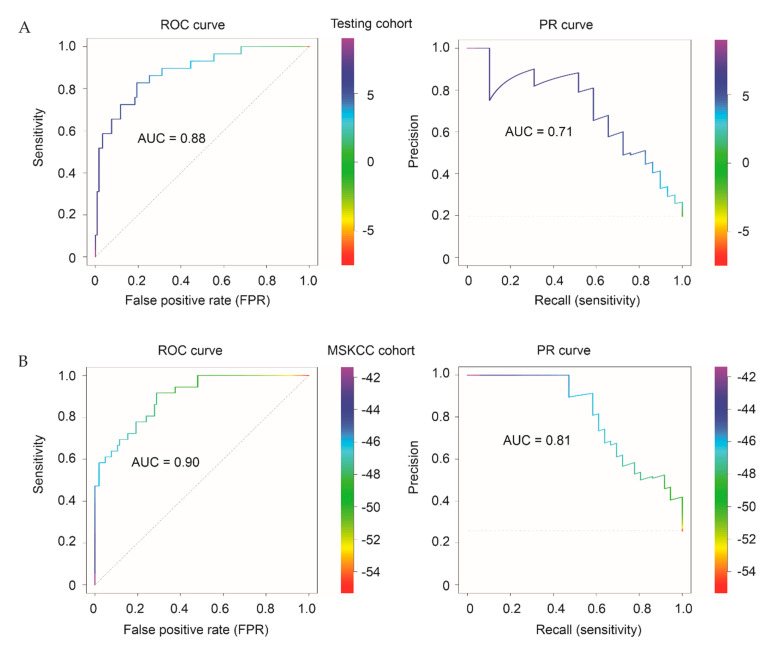
Examination of Sig27gene discriminating ability towards BCR in the Testing cohort (**A**) and an independent MSKCC dataset (**B**) using ROC and precision-recall (PR) curves. The bars on the right side show threshold ranges. The R PRROC package was used for these analyses.

**Figure 10 cancers-13-00430-f010:**
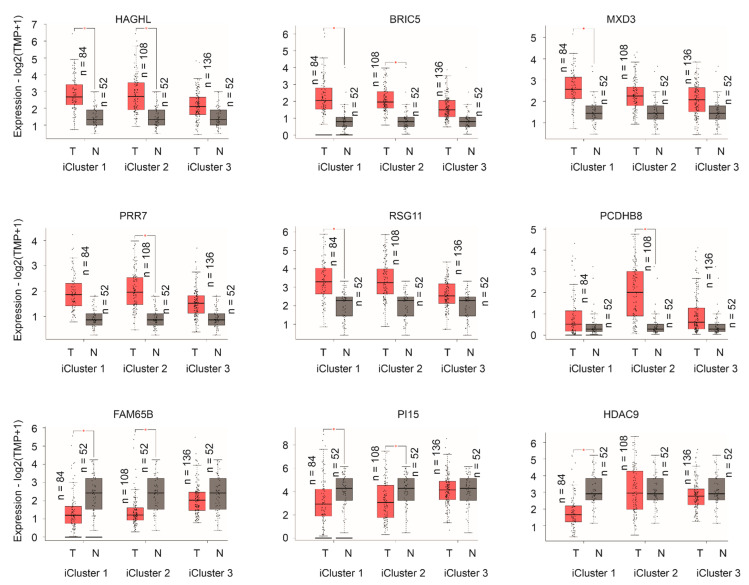
Differential expression of the indicated component genes of Sig27gene. The indicated gene expressions (mRNA) in PC (T) and matched normal prostate tissues (N) were analyzed using the GEPIA2 program. * *p* < 0.05.

**Figure 11 cancers-13-00430-f011:**
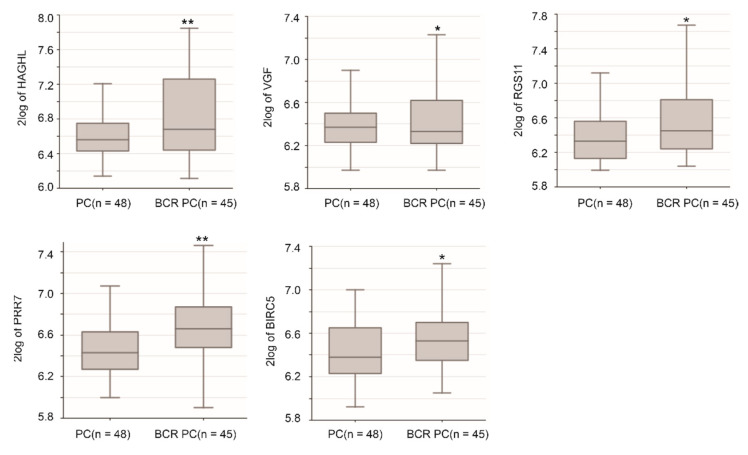
Detection of Sig27gene component gene expression in primary PCs with BCR development (BCR PC) and those without the progression (PC). Analyses were performed using the Dunning dataset in R2: Genomics Analysis and Visualization Platform. Gene expression in the dataset was determined using microarray. The expression of the indicated genes was presented as log2-transformed data. Statistical analyses were performed by the R2 Platform using one-way ANOVA. * *p* < 0.05; ** *p* < 0.01.

**Figure 12 cancers-13-00430-f012:**
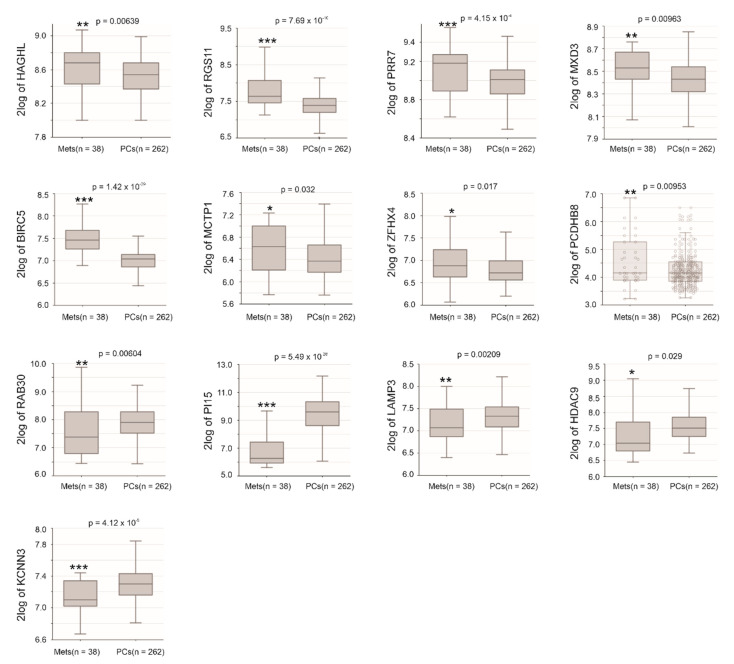
Examination of Sig27gene component gene expression in metastatic PCs (Mets) and primary PCs (PCs) using the Sawyers dataset in R2: Genomics Analysis and Visualization Platform. Gene expression in the dataset was determined using microarray. The expression of the indicated genes was presented as log2-transformed data. Statistical analyses were performed by the R2 Platform using one-way ANOVA. * *p* < 0.05; ** *p* < 0.01; *** *p* < 0.001.

**Figure 13 cancers-13-00430-f013:**
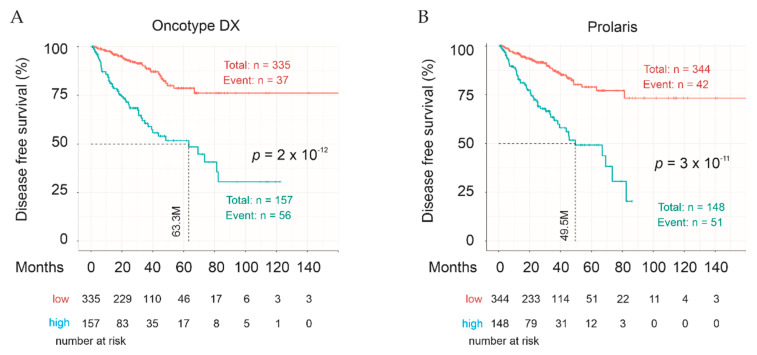
Oncotye DX- and Prolaris-mediated stratification of PC relapse risk using the TCGA PanCancer dataset. The 12 testing genes of Oncotype DX (**A**) and 31 testing genes of Prolaris (**B**) were used to calculate the panel score. Cutoff points for Oncotye DX (**A**) and Prolaris (**B**) panel scores were determined using the Maximally Selected Rank Statistics. Statistical analyses were carried out with logrank test. Kaplan Meier analyses and logrank tests were performed using the R Survival package.

**Table 1 cancers-13-00430-t001:** Human hallmark gene set enrichment of IQGAP1 DEGs.

Pathway	P.Adj	ES	NES	Size
INFγ response	0.004662	−0.53228	−2.77404	92
Apical junction	0.004662	−0.44445	−2.25531	78
Inflammatory response	0.004662	−0.58813	−3.02081	84
Il2_Stat5 signaling	0.004662	−0.4212	−2.14177	80
Complement	0.004662	−0.42784	−2.17532	81
Allograft rejection	0.004662	−0.55785	−2.83455	82
KRAS signaling UP	0.004662	−0.62089	−3.10098	73
UV response DN	0.004662	−0.52657	−2.58735	67
TNFα signaling via NFκB	0.004662	−0.56906	−2.80937	70
Epithelial mesenchymal transition	0.004662	−0.64204	−3.10811	63
Estrogen response early	0.004662	−0.50057	−2.37491	59
INFα response	0.004662	−0.47271	−2.12179	45
IL6 Jak Stat3 signaling	0.004662	−0.60951	−2.574	33
Estrogen response late	0.007924	−0.42778	−1.91355	44
Coagulation	0.007924	−0.46215	−2.02948	41
Myc targets V2	0.011409	0.481626	2.48835	32
Fatty acid metabolism	0.011897	0.30286	1.71898	46
Apoptosis	0.011897	−0.36188	−1.74708	62
Hypoxia	0.011897	−0.37568	−1.79535	60
DNA repair	0.013372	0.37213	2.35074	69
Myc targets V1	0.016632	0.41489	2.90611	102
Mitotic spindle	0.016632	−0.32065	−1.71067	103
Myogenesis	0.017405	−0.37748	−1.71929	49
Oxidative phosphorylation	0.017405	0.508354	3.58833	113
TGFβ signaling	0.017405	−0.44961	−1.80892	28

P.Adj: adjusted *p* value; ES: enrichment score; NES: normalized enrichment score.

**Table 2 cancers-13-00430-t002:** The component genes of Sig27gene.

Gene	Locus	Log2 Ratio ^1^	*p*-Value	*q*-Value
*HAGHL*	16p13.3	1.72	3.54 × 10^−17^ ***	1.00 × 10^−15^ ***
*LCN12*	9q34.3	1.34	6.75 × 10^−18^ ***	2.39 × 10^−16^ ***
*DCST2*	1q21.3	1.28	6.10 × 10^−16^ ***	1.14 × 10^−14^ ***
*VGF*	7q22.1	1.2	5.54 × 10^−8^ ***	1.53 × 10^−7^ ***
*RGS11*	16p13.3	1.18	9.68 × 10^−12^ ***	6.41 × 10^−11^ ***
*PRR7*	5q35.3	1.17	1.65 × 10^−16^ ***	3.82 × 10^−15^ ***
*LINC01089*	12q24.31	1.1	1.43 × 10^−15^ ***	2.37 × 10^−14^ ***
*MXD3*	5q35.3	1.07	1.83 × 10^−26^ ***	3.50 × 10^−23^ ***
*BIRC5*	17q25.3	1.04	1.70 × 10^−10^ ***	8.74 × 10^−10^ ***
*LTC4S*	5q35.3	1.03	1.68 × 10^−11^ ***	1.06 × 10^−10^ ***
*H1FX-AS1*	3q21.3	1.03	2.83 × 10^−17^ ***	8.46 × 10^−16^ ***
*FPR3*	19q13.41	−1	1.12 × 10^−10^ ***	5.96 × 10^−10^ ***
*RAB30*	11q14.1	−1	8.65 × 10^−12^ ***	5.79 × 10^−11^ ***
*RIPOR2*	6p22.3	−1.01	3.04 × 10^−5^ ***	6.67 × 10^−5^ ***
*NOD2*	16q12.1	−1.01	1.05 × 10^−8^ ***	3.93 × 10^−8^ ***
*PLXNA4*	7q32.3	−1.02	4.74 × 10^−11^ ***	2.71 × 10^−10^ ***
*RRAGC*	1p34.3	−1.03	1.33 × 10^−11^ ***	8.61 × 10^−11^ ***
*TF × 10C*	7q31.2	−1.06	5.65 × 10^−13^ ***	4.84 × 10^−12^ ***
*PI15*	8q21.13	−1.08	5.34 × 10^−4^ ***	9.88 × 10^−4^ ***
*ZFHX4*	8q21.13	−1.08	1.92 × 10^−8^ ***	6.90 × 10^−8^ ***
*LAMP3*	3q27.1	−1.21	7.78 × 10^−10^ ***	3.55 × 10^−9^ ***
*HDAC9*	7p21.1	−1.24	1.07 × 10^−7^ ***	3.83 × 10^−7^ ***
*MCTP1*	5q15	−1.29	1.27 × 10^−11^ ***	8.23 × 10^−11^ ***
*KCNN3*	1q21.3	−1.4	5.58 × 10^−10^ ***	2.73 × 10^−9^ ***
*PCDHB8*	5q31.3	−1.42	1.39 × 10^−8^ ***	5.09 × 10^−8^ ***
*PCDHGB2*	5q31.3	−1.66	4.86 × 10^−9^ ***	1.93 × 10^−8^ ***
*PCDHGA5*	5q31.3	−1.94	6.00 × 10^−10^ ***	2.79 × 10^−9^ ***

^1^: IQGAP1 downregulation group vs. the group without the downregulation. ***: *p* < 0.001.

**Table 3 cancers-13-00430-t003:** Univariate and multivariate Cox analysis of Sig27gene for PC DFS.

Factors	Univariate Cox Analysis			Multivariate Cox Analysis		
	HR	95% CI	*p*-Value	HR	95% CI	*p*-Value
Sig27gene	2.72	2.27–3.25	<2 × 10^−16^ ***	2.32	1.88–2.88	5.74 × 10^−15^ ***
Age ^1^	1.02	0.99–1.05	0.189	0.99	0.96–1.02	0.402
WHO IV ^2^	9.76	1.28–74.61	0.02817 *	3.46	0.43–27.83	0.244
WHO V ^2^	21.38	2.96–154.52	0.00241 **	5.11	0.66–39.64	0.199
Margin 1 ^3^	2.30	1.52–3.48	8.1 × 10^−5^ ***	1.28	0.81–2.03	0.294
Tstage 1 ^4^	3.69	2.08–6.52	7.54 × 10^−6^ ***	1.34	0.70–2.69	0.352

Analyses were performed using the TCGA PanCancer cohort. ^1^: Age at diagnosis; ^2^: WHO prostate cancer grade IV and V in comparison to WHO prostate cancer grade I; ^3^: Surgical margin 1 compared to surgical margin 0; ^4^: Tstage 1: tumor stage 1 (3 + 4) in comparison to Tstage 0 (tumor stage 1 + 2). * *p* < 0.05; ** *p* < 0.01; *** *p* < 0.001.

**Table 4 cancers-13-00430-t004:** Association of Sig27gene component genes with PC recurrence.

Gene	Details	HR ^a^	95% CI	*p*-Value
HAGHL ^c^	Hydroxyacylglutathione Hydrolase Like	1.002	1.001–1.003	3.6 × 10^−6^ ***
LCN12 ^c^	Lipocalin 12	1.009	1.004–1.015	8.83 × 10^−4^ ***
DCST2 ^c^	DC-STAMP Domain Containing 2	1.01	1.005–1.015	3.61 × 10^−5^ ***
VGF ^c^	VGF Nerve Growth Factor Inducible	1.003	1.001–1.004	4.17 × 10^−6^ ***
RGS11	Regulator of G Protein Signaling 11	1.001	1.001–1.001	4.76 × 10^−7^ ***
PRR7 ^c^	Proline Rich 7, Synaptic	1.005	1.003–1.007	1.27 × 10^−7^ ***
LINC01089 ^c^	Long Intergenic Non-Protein Coding RNA 1089	1.001	1–1.001	2.66 × 10^−5^ ***
MXD3 ^c^	MAX Dimerization Protein 3	1.004	1.002–1.005	2.05 × 10^−7^ ***
BIRC5	Baculoviral IAP Repeat Containing 5 (survivin)	1.002	1.002–1.003	2.78 × 10^−7^ ***
LTC4S ^c^	Leukotriene C4 Synthase	1.017	1.011–1.023	3.12 × 10^−8^ ***
H1FX-AS1 ^c^	H1−10 Antisense RNA 1	1.006	1.004–1.009	2.28 × 10^−7^ ***
FPR3	Formyl Peptide Receptor 3	1.002	1.001–1.004	0.00434 **
RAB30	member RAS Oncogene Family	1.001	0.998–1.004	0.444
RIPOR2	Atypical inhibitor of the small G protein RhoA	1	1–1	0.0464 *
NOD2 ^c^	Nucleotide Binding Oligomerization Domain Containing 2	1.01	1.004–1.016	9.95 × 10^−4^ ***
PLXNA4	Plexin A4	1.004	1.001–1.006	0.0177 *
RRAGC	Ras Related GTP Binding C	1	0.998–1.003	0.768
TFEC	Transcription Factor EC	1.005	1.002–1.008	0.0194 *
PI15	Peptidase Inhibitor 15	1	1–1	0.12
ZFHX4	Zinc Finger Homeobox 4	1.001	1–1.002	0.0385 *
LAMP3 ^b^	Lysosomal Associated Membrane Protein 3	1.001	1–1.001	0.0616
HDAC9	Histone Deacetylase 9	1	0.9998–1	0.585
MCTP1	Multiple C2 And Transmembrane Domain Containing 1	1.003	1–1.005	0.0497 *
KCNN3	Potassium Calcium-Activated Channel Subfamily N Member 3	1.005	0.997–1.009	0.0668
PCDHB8	Protocadherin Beta 8	1	0.9991–1.002	0.523
PCDHGB2	Protocadherin Gamma Subfamily B2	1.001	1–1.002	0.00166 **
PCDHGA5	Protocadherin Gamma Subfamily A5	1	1–1.001	0.0109 *

^a^: Determined by univariate Cox PH (proportional hazard) analysis; ^b^: PH assumption was not confirmed; ^c^: independent risk factors of PC relapse (*p* < 0.05) after adjusting for age at diagnosis, WHO prostate cancer grade, surgical margin, and tumor stage. * *p* < 0.05; ** *p* < 0.01; *** *p* < 0.001.

**Table 5 cancers-13-00430-t005:** Oncogenic role of the Sig27gene component genes.

Gene	Oncogenic Role in PC	Oncogenic Role in Others	Refs
HAGHL	unknown	unknown	NA
LCN12	unknown	unknown	NA
DCST2	unknown	unknown	NA
VGF	facilitation of radioresistance in prostate cancer cells	promotion of resistance to tyrosine kinase inhibitors in lung cancer	[60,61]
RGS11	association with TMPRSS2-ERG	a biomarker of lung cancer	[62,63]
PRR7	unknown	unknown	NA
LINC01089	unknown	suppression of cell proliferation	[64]
MXD3	unknown	promotion of medullobastoma	[65]
BIRC5	promotion of prostate cancer	promotion of cancer progression and metastasis	[66,67]
LTC4S	predicting prostate cancer progression	a component gene of a immune signature of breast cancer	[68,69]
H1FX-AS1	unknown	reverse association with cervical cancer prognosis	[70]
FPR3	unknown	sustain meiotic recombination checkpoint actions	[71]
RAB30	unknown	association of good prognosis in triple negative breast cancer	[72]
RIPOR2	unknown	association of immune cell infiltration and thus inhibition of cervical cancer	[73]
NOD2	induction of innate immune responses of prostate cancer	immunosuppression of tumorigenesis in gastric cancer	[74,75]
PLXNA4	unknown	inhibition of tumor cell migration and contribution to innate immunity in working with Toll-like receptor	[76,77]
RRAGC	unknown	regulation (activation and inactivation) of mTOC1 activation	[78]
TFEC	unknown	regulation of lysosome biogenesis and mTOR activation	[79]
PI15	methylation of its CpGs occurs in metastatic PCs	biomarker of cholangiocarcinoma	[80]
ZFHX4	unknown	a susceptibility locus of cutaneous basal cell carcinoma	[81]
LAMP3 b	evidence suggests its association with resistance to platinum in PC	a hypoxia-induced gene associated with aggressive breast cancer	[82,83]
HDAC9	mutations detected in PC	increases in expression in bladder cancer	[84,85]
MCTP1	unknown	downregulation in paclitaxel-resistant ovarian cancer cells	[86]
KCNN3	unknown	suppression of bladder cancer cell migration and invasion	[87]
PCDHB8	unknown	unknown	NA
PCDHGB2	unknown	unknown	NA
PCDHGA5	unknown	unknown	NA

NA: not available.

## Data Availability

Data is present in this article and Appendix A.

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
