# Peer review of "Effective Prediction of Prostate Cancer Recurrence through the IQGAP1 Network"

_cancers, 2021, doi:10.3390/cancers13030430_

Round 1
Reviewer 1 Report
The authors present interesting data about the role of IQ motif GTPase-activating scaffold protein 1 (IQGAP1) and a related gene signature in prostate cancer (PC). Though being mainly described as pro-oncogenic in several cancer entities the found it to be underexpressed in CRPC and advanced PCs. This was first shown in a small cohort of primary PC samples (n=26) by immunohistochemistry. Then they validated these findings in several publicly available datasets obtained from Oncomine. In these datasets they either found a lower expression of IQGAP1 in primary PC compared to normal tissue (Liu and Wallace datasets) or lower expression in distant metastases compared to primary PC (Grasso, Chandran and Sawyer datasets). IHC of prostate cancer xenografts, primary prostate cancers and tumors derived in TRAMP mice showed the expression if IQGAP1 mainly to be localized at the plasma membrane.
By in silico analyses and by analyses of a limited number of PTEN -/- mice primary tumors and LNCaP xenografts the authors could show that the expression of IQGAP1 is highly reduced in CRPC state and androgen suppression, respectively.
Afterwards they stratified the TCGA prostate cancer dataset in low and high-risk groups according to their IQGAP1 expression. Subsequently the identified 598 genes differentially expressed in those two sub-cohorts. With these differentially expressed genes (DEGs) they performed pathway cluster analysis and Metascape network analysis, which revealed several potentially deregulated pathways potentially interacting with IQGAP1. By subsequent arbitrary division of the TCGA dataset in a testing and a training dataset. By this they established, validated and refined a gene expression signature which is able to identify patients at high risk of short DFS. This could be validated in the independent MSKCC dataset even with a higher AUC. Furthermore, the showed multiple of the genes that are contained in the Sig27 gene set to be overexpressed in prostate cancer samples which belong to integrative clusters 1 and 2 but not to cluster 3. Several of these genes were also significantly higher expressed in PC with biochemical recurrence vs. PC without biochemical recurrence and an even greater number were differentially expressed between primary tumors and metastases.
In summary the authors established a 27 gene expression signature associated with short BCR and advanced tumor stage (especially metastases). Besides, some of the genes contained in this signature (e.g. MXD3 and PCDH8) have not been described in PC before. Taken together the manuscript described a detailed workup for marker panel identification in PC using patient samples, xenografts/mouse models and several in silico cohorts. Though several endeavors have been made to define and establish multi-gene biomarker panels in PC throughout the last years, due to its specific approach and the identification of genes until now not described on the context of PC, this manuscript comprises relevant novelty for the scientific community.
Yet, from my point of view there are some minor points to be addressed:
- The cohort sizes of MSKCC cohorts shown in Figure 4 F are very unbalanced (low expression of IQGAP1 = 16, high expression of IQGAP1 = 124). And also the TCGA cohort shows a comparably smaller proportion of tumors with low IQGAP1 expression (114 vs. 378). This equals to a proportion of 11.4 – 23.1%. Subsequent analyses in the TCGA cohort using the Sig27 gene signature result in fractions of 11.9 – 17.8% with a high Sig27 score and therefore being identified as having a poor prognosis. Validation in the MSKCC dataset resulted in fractions of 26.4 and 17.1%. Can other features (either clinical or known molecular features like loss of PTEN, T2:E fusions etc.) be attributed to this patient group?
- Does table 3 refer to the TCGA dataset? How were the results of multivariable analyses in MSKCC dataset? Could the serum PSA-level be included in this analysis?
- The labeling of the different Kaplan-Meier curves in Figure 7 is a bit difficult to follow. It would be helpful to label “training”, “testing” and “total” set directly above the figures and not only in the legend.
- The same would also be helpful for ROC curves in figure 9.
- The classification of integrative clusters 1-3, based on the publication of Shen et al 2009 is quite old. Can similar distribution of Sig27 gene differential expression also be associated with other classification systems, e.g. TCGA prostate cancer subtypes?
- The simple summary and the abstract contain several quite odd formulations und should be revised.
Author Response
We appreciate the reviewer’s overall positive tone and insightful remarks. Here are our detailed revisions.
“The cohort sizes of MSKCC cohorts shown in Figure 4 F are very unbalanced (low expression of IQGAP1 = 16, high expression of IQGAP1 = 124). And also the TCGA cohort shows a comparably smaller proportion of tumors with low IQGAP1 expression (114 vs. 378). This equals to a proportion of 11.4 – 23.1%. Subsequent analyses in the TCGA cohort using the Sig27 gene signature result in fractions of 11.9 – 17.8% with a high Sig27 score and therefore being identified as having a poor prognosis. Validation in the MSKCC dataset resulted in fractions of 26.4 and 17.1%. Can other features (either clinical or known molecular features like loss of PTEN, T2:E fusions etc.) be attributed to this patient group?”
Authors' response – Yes, we agree with this assessment. Clinical features such as tumor grade (Gleason scores), stage, surgical margin, and others are well-established characteristics of PC aggressiveness. The proportion of tumors with these features in a cohort is attributable to the fraction of tumors with high risk of recurrence. This was the reason to analyze Sig27gene-derived prediction of PC relapse together with these clinical features using the multivariate Cox model. As the manuscript is already long and complex, we thought it might be ok for not discussing this concept so to keep the manuscript simple. We hope the reviewer will agree with this arrangement.
“Does table 3 refer to the TCGA dataset? How were the results of multivariable analyses in MSKCC dataset? Could the serum PSA-level be included in this analysis?”
Authors' response – Yes, the TCGA PanCancer dataset was used for Table 3; a footnote was added to indicate this (line 324, marked with red). For MSKCC, these clinical features are not available. The PSA level is also not available for the TCGA PanCancer dataset.
“The labeling of the different Kaplan-Meier curves in Figure 7 is a bit difficult to follow. It would be helpful to label “training”, “testing” and “total” set directly above the figures and not only in the legend.”
“The same would also be helpful for ROC curves in figure 9.”
Authors' response – Thanks for the suggestions. Cohort identities have been added in both figures (pages 13 and 16).
“The classification of integrative clusters 1-3, based on the publication of Shen et al 2009 is quite old. Can similar distribution of Sig27 gene differential expression also be associated with other classification systems, e.g. TCGA prostate cancer subtypes?”
Authors' response – We agree with these knowledgeable comments. We trust the reviewer agrees that classification of PC is not commonly used, although there were a number of reports, like ERG fusion-based and using SPOP, FOXA, IDH1, and others. This is clearly different from breast cancer with respect to subtypes of ER+, HER2+, and triple negative. As well, the TCGA PanCancer dataset does not present other classifications or subtypes. We thus did not make efforts for these analyses. Hope the reviewer will agree our view that without it may not compromise this investigation.
“The simple summary and the abstract contain several quite odd formulations und should be revised.”
Authors' response – Thanks for pointing this out. Both have been thoroughly revised with changes marked with red.
Reviewer 2 Report
This manuscript describes that IQGAP1 is downregulated in castration resistant PC (CRPC) and advanced prostate cancers (PCs). The downregulations were associated with poor PC recurrence in the TCGA PanCancer and MSKCC cohorts. Differential expressed genes relative to IQGAP1 downregulation were identified with enrichment in chemotaxis, cytokine signaling, and others along with reductions in immune responses. A novel 27-gene signature (Sig27gene) was constructed from these differentially expressed genes (DEGs) through random division of the TCGA cohort into a Training and Testing population. The panel was validated using an independent MSKCC cohort. Sig27gene robustly predicts PC recurrence and Sig27gene independently predicts PC recurrence after adjusting for multiple clinical features. This manuscript is well studied and written.
In Abstract, “DEGs” should be “differentially expressed genes (DEGs)”.
Several publications have reported prostate cancer signature genes. Please compare Sig27gene and with the reported signature genes and discuss advantages and weak points of Sig27gene.
Author Response
We appreciate the reviewer for his/her encourage comments. Here are our detailed revisions.
“In Abstract, “DEGs” should be “differentially expressed genes (DEGs)””
Authors' response – Changed (line 30, marked with red). Thanks.
“Several publications have reported prostate cancer signature genes. Please compare Sig27gene and with the reported signature genes and discuss advantages and weak points of Sig27gene.”
Authors' response – We thank the reviewer for this insightful remark. Yes, there are numerous publications of this type. We thought the most well-studied multigene panels being Oncotype DX, Prolaris (cell cycle progression/CCP), and Decipher. The 17-gene Oncotype DX and 31-gene Prolaris are for PC recurrence (BCR) risk assessment following radical prostatectomy (RP); the 22-gene Decipher is for predicting the risk of metastasis after RP. We thus analyzed Sig27gene together with Oncotype DX and Prolaris; a new figure (Figure 13, page 21) was prepared and discussed (lines 381-393, marked with red). Interestingly, Sig27gene seems overperforming both Oncotype DX and Prolaris in the stratification of PC relapse risk. Features of Oncotype DX, Prolaris, and Sig27gene were discussed along with their potential applications (lines 475-481, marked with red). This addition enhances this manuscript, for which we thank the reviewer for the comments.
Reviewer 3 Report
In the current MS, authors have analyzed the IQGAP1 expression in 1) published data sets of primary prostate cancer 2) in prostate cancer cells xenografts, 3) PCs Tumor of PTEN null and TRAMP mice, and 4) CRPC produced by LNCaP xenografts and PTEN null mice and observed IQGAP1 is significantly downregulated in CRPC and advanced PCs. The study is designed and executed well but there are several concerns need to be addressed-
- In the abstracts author should correct the numbering for different combinations. And throughout the MS English language editing is required e.g. ‘PCs of PTEN-/- and TRAMP mice should be corrected as tumor of PTEN-/- and TRAMP”; In abstract 27-gene signature abbreviated as (Sig27gene) while other places it is “Twenty-seven DEGs (Sig27gene)” that needs to be unified along with others.
- The downregulation of IQGAP1 expression in advanced PCa, indicates that it is a onco-suppressor in PC, while authors wrote IQGAP1 is pro-oncogenic.
- A sentence in abstract: The downregulations were associated with poor PC recurrence shows a contradictory conclusion. While if it downregulated then recurrence process should be rapid.
- How many human PC tissues were used for this study obtained from Hamilton Health Sciences, Hamilton, Ontario, Canada, I hope it is n=14 and n=13. Also, for clarity, it should be mentioned clearly that (as in abstract n=1100) if it is published study or primary PCa tissue sample used by authors?
- The Figure legends should be self-explanatory. In legend of figure 1, authors should mentioned the quantification method used e.g. ImageJ or other and how background was normalized? And also used quantification subsequent IHC.
- “PCs produced in TRAMP transgenic mice” the PCs is not a appropriates term. it should be “Tumor”
- For validation using mice model, author should also check the IQGAP1 level at gene and proteins level in primary tumor tissue, mice xenograft using qRT-PCR and Western blot.
Author Response
Reviewer’s remarks were generally positive and encouraging. Here are our detailed revisions.
“In the abstracts author should correct the numbering for different combinations. And throughout the MS English language editing is required e.g. ‘PCs of PTEN-/- and TRAMP mice should be corrected as tumor of PTEN-/- and TRAMP”; In abstract 27-gene signature abbreviated as (Sig27gene) while other places it is “Twenty-seven DEGs (Sig27gene)” that needs to be unified along with others.”
Authors' response – We thank the reviewer’s emphasis for precision. This type of errors has been corrected in the Abstract and throughout the manuscript (lines 26, 111, 132, 225; marked with red).
“The downregulation of IQGAP1 expression in advanced PCa, indicates that it is a onco-suppressor in PC, while authors wrote IQGAP1 is pro-oncogenic.”
Authors' response – We agree with descriptions being conflicting, which was essentially attributable to both pro-oncogenic and anti-oncogenic effects being reported for IQGAP1. In this revision, we have tried to avoid using of “pro-oncogenic” for IQGAP1 at least in PC. For instance, this type of description was removed from the beginning of the “Results” section (line 86).
“A sentence in abstract: The downregulations were associated with poor PC recurrence shows a contradictory conclusion. While if it downregulated then recurrence process should be rapid.”
Authors' response – Thanks for pointing out the inconsistence. The error has been corrected (Abstract, line 29, marked with red).
“How many human PC tissues were used for this study obtained from Hamilton Health Sciences, Hamilton, Ontario, Canada, I hope it is n=14 and n=13. Also, for clarity, it should be mentioned clearly that (as in abstract n=1100) if it is published study or primary PCa tissue sample used by authors?”
Authors' response – We confirmed the sample size from our hospital. The tissues used in this research were new, i.e. not been used in our previous publications. Clarification for this was added (line 90, marked with red).
“The Figure legends should be self-explanatory. In legend of figure 1, authors should mentioned the quantification method used e.g. ImageJ or other and how background was normalized? And also used quantification subsequent IHC.”
Authors' response – As requested, these details are included in the legends of figures 1 (lines 97-99, marked with red), 3 (lines 133-135, marked with red), and 4 (lines 168-171, marked with red).
““PCs produced in TRAMP transgenic mice” the PCs is not a appropriates term. it should be “Tumor””
Authors' response – Corrected (Figure 3 legend, line 132, marked with red).
“For validation using mice model, author should also check the IQGAP1 level at gene and proteins level in primary tumor tissue, mice xenograft using qRT-PCR and Western blot.”
Authors' response – We see reviewer’s emphasis on using multiple horizontal approaches. However, there are limitations to use them all under certain situations. This may apply here with respect to the use of real-time PCR and Western blot approaches. 1) The materials for tumor tissues produced in PTEN-/- mice were limited and 2) under the current COVID-19 restrictions, University is under lockdown and non-essential research is generally not permitted.
Nonetheless, we agree with the Reviewer that inclusion of this data will be helpful. As such, qRT-PCR was performed to examine the mRNA alterations of IQGAP1 in LNCaP xenografts between the intact and CRPC group. The IQGAP1 mRNA expression is significantly reduced in CRPC group compared to androgen sensitive group in the LNCaP xenograft model, validating the downregulation pattern observed in protein expression analyzed via IHC. This data was discussed (page 6, line 145, marked with red) and presented in a new panel Figure 4E (page 7); a methodological paragraph has been added in 4.6 (page 26, lines 598-604, marked with red). We hope the reviewer will agree with this arrangement.
Reviewer 4 Report
Sir,
I have recently reviewed the manuscript "Effective prediction of prostate cancer recurrence through the IQGAP1 network" submitted by Yan Gu and co-workers to Cancers (MDPI).
These authors decided to study prostate cancer, which is indeed a major health issue globally. In this manuscript, the authors described their experience with a multigene panel consisting of 27 genes (Sig27gene) designed for predictive oncology purposes with an emphasis on prostate cancer. I believe that this approach is innovative, and the data presented here are rather robust. However, I must also admit that this will very likely require time before such approach could be completely incorporated into routine care.
Lines 85-89: this text would more likely fit into Introduction. It is not a result.
Line 92 - "All PC pathologies were confirmed by hospital pathologists" - the diagnosis in low-grade PCs was made by a single pathologist or second reading was done in some/all cases? Please, clarify this issue.
Section 4.3 - method of euthanasia is missing. Please, provide details.
Section 4.5 - Fixation protocol details are missing. Please, verify that FFPE routine procedures were applied. The protocol for negative control is of inferior quality! Isotype controls are not useless! Fortunately, the detection system from Vector (used her) is relatively reliable in my own experience. However, this is a great shortcoming of this study.
Line 550 - 551: staining intensity was quantified as HScores - please, provide a reference.
Concerning Figure 1 - why 2-tailed Student’s t-test was applied? How did you test the normality of distribution of these IHC data? Similar consideration must be done with Fig. 4D. In my eyes, nonparametric tests would be more appropriate under given circumstances. Anyway, IHC statistics require also a far better description in this manuscript - what was presented in section 4.14 seems to be insufficient for this purpose.
Concerning Figure 3 - it is critical to present a negative control here! As mentioned above, the negative controls were not performed according to the best standards. The quality of IHC here does not leave a strong and trustworthy impression. Panel A seems to be troublesome; however, in Fig3C, I can see some "hopeful" difference between tumour parenchyma and tumour stroma. Anyway, this is urgently asking for improvement.
Concerning Figure 2 - I am sorry to say that this figure is not really easy to follow. I must agree that this data mining from different databases might be rather challenging. However, I do not really believe that the informative value of this Figure for readers is somewhat great. Maybe, better colour-coding (normal prostate tissues vs primary PCs vs metastatic PCs) would be more eye-catching and more appreciated by the novice readers.
To the Discussion/Conclusion: Authors absolutely correctly stated on the numerous biomarkers and predictive tools/systems used so far and I quote "the current capacity in predicting BCR is clearly not sufficient". Indeed. It is a very great problem. However, from this point, the Discussion is a bit shallow in terms of practicality of their proposal (Sig27gene). I believe that it is necessary to discuss also the clinical application here. If Sig27gene predicts PC relapse with a high level of certainty, how can a meaningful clinical test (using Sig27gene) be incorporated into clinical care? At which stage it could serve best? BCR is a relatively late event. Should we use SIG27gene at primary PC stage? Or at mPC? Please, try to suggest/comment on optimal timing.
In conclusion, I believe that the presented manuscript is truly interesting and possesses a good potential to influence also clinical oncologists. It is using several well-established techniques in combination with robust bioinformatics. However, I believe that there are some necessary corrections still needed. Anyway, I am very keen to see the perfected version as soon as possible to support it for final publication in Cancers.
Author Response
We appreciate reviewers overall positive evaluation and agree that it will take much more work to develop Sig27gene into potential clinical applications. Here are our detailed revisions.
“Lines 85-89: this text would more likely fit into Introduction. It is not a result.”
Authors' response – Thanks for the comments. These lines were removed (see line 86 in this revision.
“Line 92 - "All PC pathologies were confirmed by hospital pathologists" - the diagnosis in low-grade PCs was made by a single pathologist or second reading was done in some/all cases? Please, clarify this issue.”
Authors' response – We have clarified this unclearness (lines 88-89, marker with red).
“Section 4.3 - method of euthanasia is missing. Please, provide details.”
Authors' response – Methodological details on euthanasia are provided (page 25, line 558, marked with red).
“Section 4.5 - Fixation protocol details are missing. Please, verify that FFPE routine procedures were applied. The protocol for negative control is of inferior quality! Isotype controls are not useless! Fortunately, the detection system from Vector (used her) is relatively reliable in my own experience. However, this is a great shortcoming of this study.”
Authors' response – All tissues were prepared following FFPE procedures; this information has been added (page 26, section 4.5, lines 582-583, marked with red). We appreciate the reviewer’s comment on the use of negative controls. We trust the reviewer will agree the common use of secondary antibody as a negative control in IHC procedure, although this is not perfect. Negative controls are added (see our response to “Concerning Figure 3 … …” below).
“Line 550 - 551: staining intensity was quantified as HScores - please, provide a reference.”
Authors' response – References for HScore were added (line 592, marked with red).
“Concerning Figure 1 - why 2-tailed Student’s t-test was applied? How did you test the normality of distribution of these IHC data? Similar consideration must be done with Fig. 4D. In my eyes, nonparametric tests would be more appropriate under given circumstances. Anyway, IHC statistics require also a far better description in this manuscript - what was presented in section 4.14 seems to be insufficient for this purpose.”
Authors' response – We thank the reviewer for these comments. Normal distributions were tested. For Figure 1, normal Gaussian normality test was performed to test for normal distribution and this dataset passed D’Agostino-Pearson normality test, Anderson-Darling test and Shapiro-Wilk test at alpha = 0.05. This indicates that this data is not inconsistent with a Gaussian distribution. Nonetheless, considering this dataset’s small size and normality test may not have adequate power to detect modest deviation from the Gaussian ideal, we further performed non-parametric two-tailed Mann-Whitney test, the p value for differences between low GS and high GS group remains significant (p = 0.0075); this test was used to analyze the data (line 96-97, marked with red). For Figure 4, the sample were too small for Anderson-Darling test and D’Agostino & Pearson test, but passed the Shapiro-Wilk normality test. For reason mentioned above, two-tailed Mann-Whitney test was performed and p value was found to be 0.0043. As this dataset passed the Shapiro-Wilk normality test, 2-tailed Student’s test was still used. The confirmation of the normal distribution is indicated in Figure 4 legend (lines 167-168, marked with red). More details of statistical analyses are provided (see page 28, section 4.14, lines 658-660, marked with red). However, we trust the reviewer agrees that most statistical analyses are currently carried out using programs, which contributed to less procedure details. Nonetheless, we made effort to provide details and trust the reviewer will see our efforts.
“Concerning Figure 3 - it is critical to present a negative control here! As mentioned above, the negative controls were not performed according to the best standards. The quality of IHC here does not leave a strong and trustworthy impression. Panel A seems to be troublesome; however, in Fig3C, I can see some "hopeful" difference between tumour parenchyma and tumour stroma. Anyway, this is urgently asking for improvement.”
Authors' response – Images showing secondary antibody as negative controls have been added (see Figure 3A).
“Concerning Figure 2 - I am sorry to say that this figure is not really easy to follow. I must agree that this data mining from different databases might be rather challenging. However, I do not really believe that the informative value of this Figure for readers is somewhat great. Maybe, better colour-coding (normal prostate tissues vs primary PCs vs metastatic PCs) would be more eye-catching and more appreciated by the novice readers.”
Authors' response – Figure 2 was arranged with different colors for unique groups of samples. We agree this makes the presentation clearer. We thank the reviewer for the comments.
“To the Discussion/Conclusion: Authors absolutely correctly stated on the numerous biomarkers and predictive tools/systems used so far and I quote "the current capacity in predicting BCR is clearly not sufficient". Indeed. It is a very great problem. However, from this point, the Discussion is a bit shallow in terms of practicality of their proposal (Sig27gene). I believe that it is necessary to discuss also the clinical application here. If Sig27gene predicts PC relapse with a high level of certainty, how can a meaningful clinical test (using Sig27gene) be incorporated into clinical care? At which stage it could serve best? BCR is a relatively late event. Should we use SIG27gene at primary PC stage? Or at mPC? Please, try to suggest/comment on optimal timing.”
Authors' response – We appreciate these thoughtful remarks. Potential applications of Sig27gene are discussed (lines 469-481, marked with red). The addition enhances the manuscripts for which we thank the reviewer for the comments.
Round 2
Reviewer 3 Report
In the abstract the numbering of different combinations should be corrected-
Abstract: IQGAP1 expression was analyzed in 1) primary prostate cancer (PC, n > 1100),
2) xenografts produced from LNCaP, DU145, and PC3 cells,
3) tumor of PTEN-/- and TRAMP mice, and
3) castration resistant PC (CRPC) produced by LNCaP xenografts and PTEN-/- mice
Author Response
We appreciate reviewer's remarks. Here are our responses
“In the abstract the numbering of different combinations should be corrected- Abstract: IQGAP1 expression was analyzed in 1) primary prostate cancer (PC, n > 1100), 2) xenografts produced from LNCaP, DU145, and PC3 cells, 3) tumor of PTEN-/- and TRAMP mice, and 3) castration resistant PC (CRPC) produced by LNCaP xenografts and PTEN-/- mice”
Authors' response – Our understanding on Reviewer’s remarks is to make these statements in Abstract consistent with respect to the number of primary tumors (patients-derived) and different xenografts used. Adding details on the number of xenografts used may make Abstract too details without compromising other important messages; this is likely to occur considering the limited space available (200 words and Cancers style of n = x but not n=x). More importantly, all relevant mouse numbers involved are in the respective figure legends. It will be redundant to repeat them in Abstract. Considering this and the reviewer’s request for consistency, we have removed “(PC, n > 1100)” from Abstract (see line 25); the number of primary tumors used can be interpreted from the following statements in Abstract.
In meanwhile, we realized the missing of the number of TRAMP mice used in Figure 3 legend; n = 5 was added (line 131, marked with red). For which, we thank Reviewer #3 for this request.
Reviewer 4 Report
Sir,
I have recently reviewed the manuscript "Effective prediction of prostate cancer recurrence through the IQGAP1 network" - (version 2).
I have carefully studied rebuttal letter and confronted with the modified version 2 text.
Authors´ replies are honest and straightforward, which I greatly appreciate. There ale no unsolved issues now.
As of now, I believe that the text reached significant improvement in many ways. Moreover, I believe that it is an important contribution to the field now, and it will be greatly attractive to readers.
Therefore, I am fully supporting this outstanding manuscript for publication.
Author Response
We appreciate reviewer’s warm comments and support.